# Emergent superconductivity in topological-kagome-magnet/metal heterostructures

He Wang[1,2,8], Yanzhao Liu[1,8], Ming Gong [1,8], Hua Jiang [3,8], Xiaoyue Gao[1], Wenlong Ma [1], Jiawei Luo[1], Haoran Ji[1], Jun Ge[1], Shuang Jia [1], Peng Gao [1], Ziqiang Wang [4] ✉, X. C. Xie[1,5,6] & Jian Wang [1,5,7] ✉

Itinerant kagome lattice magnets exhibit many novel correlated and topological quantum electronic states with broken time-reversal symmetry. Superconductivity, however, has not been observed in this class of materials, presenting a roadblock in a promising path toward topological superconductivity. Here, we report that novel superconductivity can emerge at the interface of kagome Chern magnet $TbMn_6Sn_6$ and metal heterostructures when elemental metallic thin films are deposited on either the top (001) surface or the side surfaces. Superconductivity is also successfully induced and systematically studied by using various types of metallic tips on different $TbMn_6Sn_6$ surfaces in point-contact measurements. The anisotropy of the superconducting upper critical field suggests that the emergent superconductivity is quasi-two-dimensional. Remarkably, the interface superconductor couples to the magnetic order of the kagome metal and exhibits a hysteretic magnetoresistance in the superconducting states. Taking into account the spin-orbit coupling, the observed interface superconductivity can be a surprising and more realistic realization of the $p$-wave topological superconductors theoretically proposed for two-dimensional semiconductors proximity-coupled to $s$-wave superconductors and insulating ferromagnets. Our findings of robust superconductivity in topological-Chern-magnet/metal heterostructures offer a new direction for investigating spin-triplet pairing and topological superconductivity.

The heterostructure interface between two different materials has become a frontier to generate and investigate emergent quantum phases[1–12] such as the quantum Hall effect[12] and novel superconductivity[1–11]. In the heterostructures formed by topological materials, the superconducting state may inherit topological properties and show great potential for non-Abelian defect excitations for topological fault-tolerant quantum computation[2,13–17].

The kagome magnet $TbMn_6Sn_6$ has recently been discovered to have Chern-gapped Dirac fermions with dissipationless chiral edge states, coexisting with ferrimagnetism below 423 K[18]. Thus, the long-sought-after time-reversal symmetry (TRS) broken spin-triplet superconducting state[19–22] and topological superconductivity[14,15,23–27] would potentially be realized if $TbMn_6Sn_6$ exhibited a superconducting ground state. However,

[1]International Center for Quantum Materials, School of Physics, Peking University, Beijing 100871, China. [2]Center for Quantum Physics and Intelligent Sciences, Department of Physics, Capital Normal University, Beijing 100048, China. [3]Institute for Advanced Study, Soochow University, Suzhou 215006, China. [4]Department of Physics, Boston College, Chestnut Hill, MA 02467, USA. [5]Hefei National Laboratory, Hefei 230088, China. [6]Institute for Nanoelectronic Devices and Quantum Computing, Fudan University, Shanghai 200433, China. [7]Collaborative Innovation Center of Quantum Matter, Beijing 100871, China. [8]These authors contributed equally: He Wang, Yanzhao Liu, Ming Gong, Hua Jiang. ✉e-mail: wangzi@bc.edu; jianwangphysics@pku.edu.cn

superconductivity has so far not been observed in TbMn$_6$Sn$_6$ and other itinerant kagome magnets.

In this work, we report the emergent superconductivity at the interface between TbMn$_6$Sn$_6$ and non-superconducting metals. After depositing metallic films (paramagnetic metal: Au and Ag; ferromagnetic metal: Ni) on the surfaces of TbMn$_6$Sn$_6$ single crystals, superconductivity is observed by standard transport measurements. The hysteretic behavior in the superconducting state suggests that the superconductivity is coupled with the magnetization of TbMn$_6$Sn$_6$. Furthermore, by pressing metallic tips (paramagnetic tips: PtIr, Au, Ag; ferromagnetic tip: Ni) onto the crystal surfaces of the samples, a superconducting phase inheriting the magnetic properties of TbMn$_6$Sn$_6$ is also induced and characterized by point-contact spectra (PCS). The structural and elemental mappings show the TbMn$_6$Sn$_6$ sample near the interface has a naturally formed degraded layer possessing polycrystalline TbMn$_6$Sn$_6$. Since TbMn$_6$Sn$_6$ is a magnetic Chern metal[18], the broken TRS and emergent superconducting states at the interface make the TbMn$_6$Sn$_6$/metal heterostructure an ideal system to realize topological superconductivity[26,27], which we demonstrate using a simple theoretical model.

## Results

### The transport and magnetic properties of TbMn$_6$Sn$_6$ samples

The high-quality TbMn$_6$Sn$_6$ single crystals are grown via a flux method[18,28]. The TbMn$_6$Sn$_6$ single crystals crystallize in a hexagonal structure with space group P6/mmm (see Supplementary Fig. 1) and show ferrimagnetic behavior below an ordering temperature ≈ 423 K[18,28]. The atomic and magnetic structures are shown in Fig. 1a. In a single unit cell, Mn atoms with out-of-plane magnetic order form two

kagome layers, and one Tb atom with opposite magnetic moment locates between the two Mn layers. The temperature dependence of the normalized longitudinal resistance ($R/R_{6K}$) of the TbMn$_6$Sn$_6$ sample is shown in Fig. 1b. The $R/R_{6K}$-$T$ curve exhibits a typically metallic behavior, and no extra features are detected at low temperatures (inset of Fig. 1b), consistent with early work[18]. The magnetic properties of TbMn$_6$Sn$_6$ are investigated by the electrical transport and magnetization measurements. As shown in Fig. 1c, the magnetization as a function of the magnetic field exhibits explicit hysteresis loops with a sizeable coercive field of about 1.8 T (at 2 K) when the magnetic field is applied along the $c$ axis (perpendicular to the kagome layer). However, under the in-plane magnetic field, the magnetization shows linear magnetic field dependence (inset of Fig. 1c). The magnetization results show that the TbMn$_6$Sn$_6$ crystal has out-of-plane magnetic order, which is consistent with the previous report[18]. The anomalous Hall effect has also been detected by transport measurements (Fig. 1d).

### The emergent superconductivity in TbMn$_6$Sn$_6$/metallic film heterostructures revealed by transport measurements

Surprisingly, when the (001) surface of the sample is capped with a 10 nm Au film, a significant resistance drop down to nearly 60% of $R_{6K}$ is observed at around 3.6 K (Fig. 2a). The drop in $R/R_{6K}$-$T$ curves is reminiscent of the superconducting transition. When the magnetic field is applied along the $c$ axis, this drop weakens and finally disappears, confirming the emergence of superconductivity at the interface between the Au film and (001) surface of TbMn$_6$Sn$_6$. Even the magnetic Ni film deposited on the (001) surface of the sample shows superconducting signals with zero-field onset $T_c$ ≈ 3.5 K (see Supplementary Fig. 2), close to that of heterostructures capped with the Au

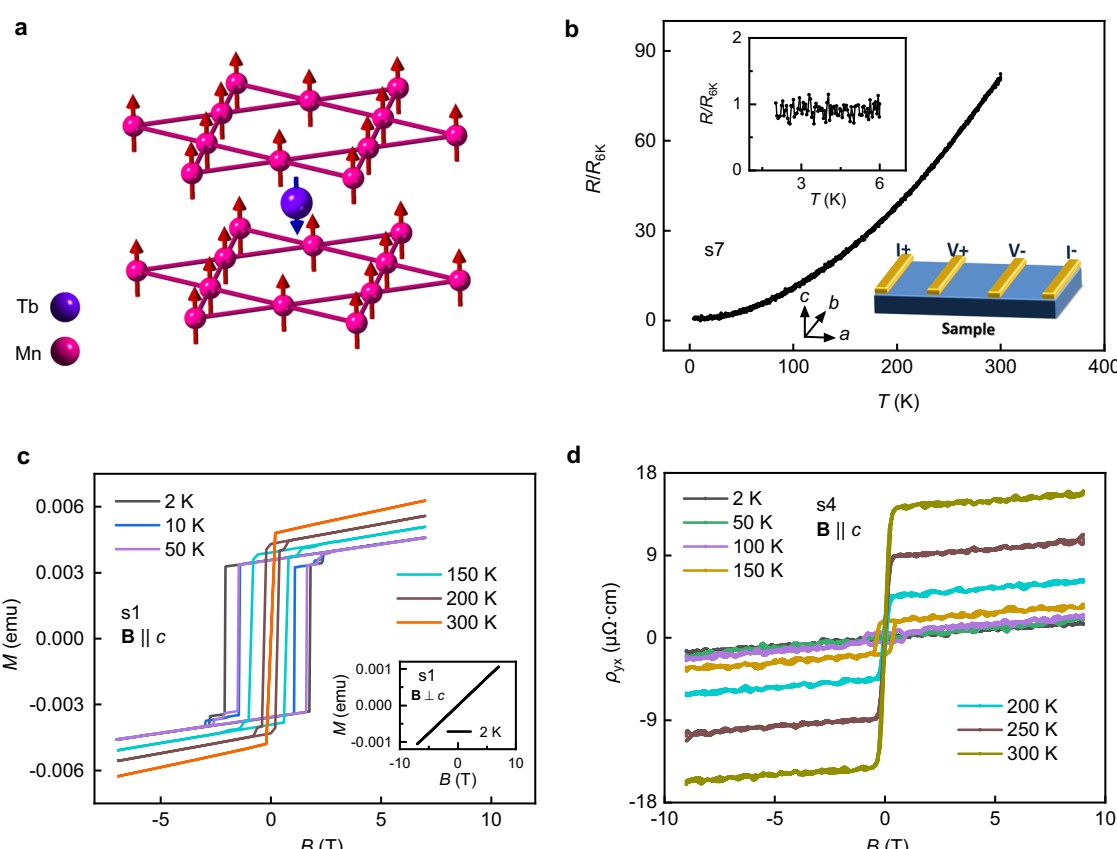

**Fig. 1 | Transport and magnetic properties of TbMn$_6$Sn$_6$ samples. a** The schematic of the magnetic structure of manganese (pink) and terbium (purple) atoms in TbMn$_6$Sn$_6$. **b** The temperature dependence of normalized resistance $R/R_{6K}$. Related resistance is measured by the standard four-electrode method. Upper inset: the zoom-in of $R/R_{6K}$ - $T$ curve below $T$ < 6 K. Lower inset: the schematic of the standard four-electrode configuration. **c** The out-of-plane (**B**∥$c$) magnetization curves at different temperatures. Inset: the in-plane (**B**⊥$c$) magnetization curve taken at 2 K. **d** The magnetic field (**B**∥$c$) evolution of Hall resistivity at different temperatures.

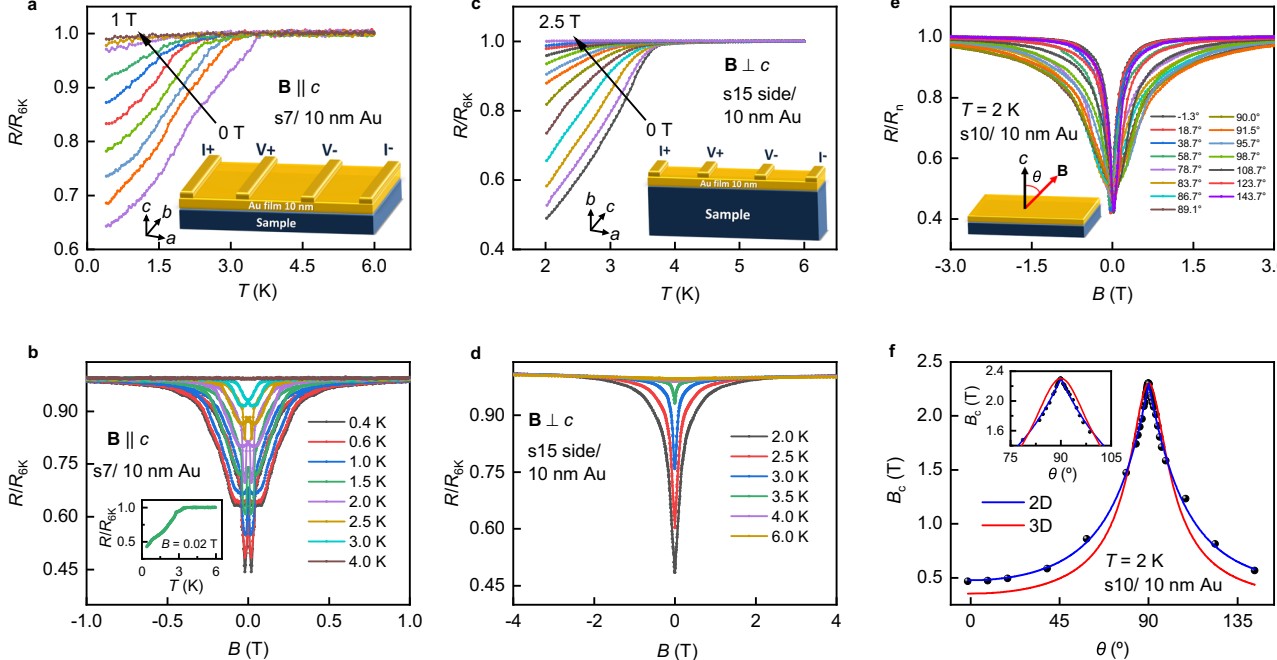

**Fig. 2 | Emergent superconductivity at the interface between the TbMn$_6$Sn$_6$ sample and the Au film. a** The evidence of superconductivity for Au film-coated (001) surface of TbMn$_6$Sn$_6$ single-crystal (s7) detected by the standard four-electrode method. The curves are measured at **B** = 0 T, 0.12 T, 0.145 T, 0.175 T, 0.21 T, 0.25 T, 0.3 T, 0.45 T, 0.7 T, and 1 T, respectively. The magnetic field is applied along the out-of-plane direction (**B** ∥ $c$ axis). Inset: the schematic of the standard four-electrode measurements for the TbMn$_6$Sn$_6$/Au film sample. **b** The magnetoresistance measurements of s7 capped with 10 nm Au film at different temperatures (**B** ∥ $c$ axis). Inset: the temperature dependence of normalized resistance at 0.02 T with a drop down to nearly 40% of $R_{6K}$. **c** The evidence of superconductivity for Au film-coated side surface of a TbMn$_6$Sn$_6$ single-crystal (s15) detected by the standard four-electrode method. The curves are measured at 0 T, 0.02 T, 0.05 T, 0.09 T, 0.14 T, 0.23 T, 0.37 T, 0.48 T, 0.65 T, 0.85 T, 1.2 T, 1.5 T, and

2.5 T, respectively. The magnetic field is applied perpendicular to the side surface (**B** ⊥ $c$ axis). Inset: the schematic of the standard four-electrode measurements for the TbMn$_6$Sn$_6$/Au film sample. **d** The magnetoresistance measurements of s15 capped with 10 nm Au film on the side surface at different temperatures (**B** ⊥ $c$ axis). **e** The angular dependent magnetoresistance curves of a TbMn$_6$Sn$_6$ sample (s10) capped with 10 nm Au film at 2 K. Here $\theta$ is the angle between the magnetic field and $c$-axis of the TbMn$_6$Sn$_6$ sample. $R_n$ is the magnetoresistance at 3 T when **B** ∥ $c$ axis. **f** Angular dependence of the critical magnetic field $B_c(\theta)$ of s10 capped with 10 nm Au film. Inset shows the zoom-in of the $B_c(\theta)$ around 90°. The blue curve is the fitting with two-dimensional (2D) Tinkham model ($(B_c(\theta) \sin(\theta)/B_{c//})^2 + |B_c(\theta)\cos(\theta)/B_{c\perp}| = 1$ and the red curve is the fitting with the three-dimensional (3D) anisotropic mass model $B_c(\theta) = B_{c//}/(\sin^2(\theta)+\gamma^2\cos^2(\theta))^{1/2}$ with $\gamma = B_{c//}/B_{c\perp}$. $B_c$ is defined as the magnetic field corresponding to 95% normal resistance.

film. The observation of a similar superconducting state for both the magnetic Ni and the nonmagnetic Au capping films on the ferrimagnet TbMn$_6$Sn$_6$ substrate suggests that TRS is broken in the interface superconductivity.

The nature of the interface superconductivity is further studied by transport measurements when the side surface of the TbMn$_6$Sn$_6$ crystal is capped with a 10 nm Au film. Figure 2c shows one set of typical results obtained from a sample labeled s15. At zero magnetic field, the resistance drops down to less than 50% of $R_{6K}$, which can be suppressed by external magnetic fields, showing the signature of superconductivity. To probe the interplay between magnetism and superconductivity, we characterize the magnetic anisotropy of TbMn$_6$Sn$_6$/Au. For the heterostructure formed by depositing the Au film on the (001) top surface of TbMn$_6$Sn$_6$, the magnetoresistance (MR) curves show hysteretic behavior when the magnetic field is perpendicular to the Au film (Fig. 2b, **B** ∥ $c$ axis). However, the hysteresis in MR curves disappears when the Au film is capped on the side surface of TbMn$_6$Sn$_6$ (Fig. 2d) and the magnetic field is applied perpendicularly to the side surface (**B** ⊥ $c$ axis). Remarkably, the hysteresis anisotropy observed in the interface superconducting phase of TbMn$_6$Sn$_6$/Au film heterostructure qualitatively follows that of the anisotropic magnetization in the non-superconducting bulk ferrimagnet TbMn$_6$Sn$_6$ (Fig. 1c). In addition, the resistance drop becomes more notable after the magnetization treatment on the TbMn$_6$Sn$_6$/Au heterostructure (Supplementary Fig. 3), indicating that magnetism is favorable for the formation of superconductivity. The ferromagnetic exchange coupling is known to frustrate the time-reversal invariant spin-singlet

Cooper pairs and promote Cooper pairs having spin-triplet pairing symmetry[29]. It is thus natural to expect that the emergent interface superconductivity in the TbMn$_6$Sn$_6$/Au film heterostructure has at least a significant triplet-pairing (such as $p$-wave) component.

Furthermore, the anisotropy of the emergent superconductivity is revealed by angular-dependent transport measurements. As shown in Fig. 2e–f, a cusp-like peak is clearly observed in the angular dependence of the critical magnetic field $B_c(\theta)$ of TbMn$_6$Sn$_6$/Au heterostructure (s10), here $\theta$ is the angle between the magnetic field and $c$-axis of TbMn$_6$Sn$_6$ sample. The peak feature can be well fitted by the two-dimensional (2D) Tinkham model and deviates from the three-dimensional (3D) anisotropic mass model[30], demonstrating quasi-2D superconductivity at the TbMn$_6$Sn$_6$/Au heterostructure. Moreover, the temperature dependence of the critical field, presented in Supplementary Figs. 4a and 4b, show the $(T_c-T)^{1/2}$-dependence of $B_{c//}(T)$ (magnetic field applied perpendicular to $c$-axis) and $T$-linear dependence of $B_{c\perp}(T)$ (magnetic field applied along the $c$-axis) near $T_c$, respectively, which are also consistent with the phenomenological 2D Ginzburg-Landau (GL) model[31]. The quasi-2D superconductivity of the TbMn$_6$Sn$_6$/Au heterostructure further supports that the observed superconductivity is emergent at the interface between TbMn$_6$Sn$_6$ and deposited metal film.

To probe the robustness of the observed interface superconductivity, the (001) top surface of samples is coated with silver epoxy, and transport properties are measured in an applied magnetic field. As shown in Supplementary Fig. 5, the results of TbMn$_6$Sn$_6$/silver epoxy confirm the robustness of the emergent interface

superconductivity between two non-superconducting partners, the topological Chern magnet and metallic films. A control experiment is further carried out to test the influence of the interface condition on the emergent superconductivity. The silver film is evaporated on $TbMn_6Sn_6$ by sputtering to form a $TbMn_6Sn_6$/Ag film heterostructure with a better interface than the $TbMn_6Sn_6$/silver epoxy heterostructure. As shown in Supplementary Fig. 6, the observed superconducting signal in $TbMn_6Sn_6$/Ag film heterostructure is more notable than that induced by the silver epoxy, indicating that good contact between the metallic film and $TbMn_6Sn_6$ is conducive to the formation of superconductivity. Due to the non-van der Waals nature of $TbMn_6Sn_6$ crystals, it is challenging to obtain a large area of atomic-level flat surfaces from the micrometer-scale $TbMn_6Sn_6$ single crystals. As a result, when depositing a metallic film onto the $TbMn_6Sn_6$ surface, it is difficult to achieve a uniform and consistent contact condition, which may lead to inhomogeneous superconductivity with non-zero residual resistance at low temperatures.

## The emergent superconductivity in $TbMn_6Sn_6$/metal heterostructures confirmed by point-contact measurements

To further investigate the emergent superconductivity at the $TbMn_6Sn_6$/metal heterostructure, different metallic tips are used to carry out the point-contact (PC) measurements on $TbMn_6Sn_6$ samples in a cryogenic system. The main results of one typical PC state obtained by pressing a PtIr tip onto the (001) surface of the sample (s1) are shown in Fig. 3. Figure 3a shows the temperature dependence of the normalized PC resistance ($R/R_{5K}$) with and without applying the magnetic field. The differential PC resistance ($R$) is obtained by using

the standard lock-in technique in a quasi-four-electrode PC configuration (the inset of Fig. 3a). When the external magnetic field is zero, a resistance drop with a transition temperature of 3.4 K (blue curve) is observed, consistent with the standard four-electrode transport measurements for the superconducting $TbMn_6Sn_6$/Au film heterostructures (Fig. 2a). This PC resistance versus temperature curve seems to undergo a two-step transition, which may attribute to the different superconducting phases triggered by complex interfacial conditions in the contact region. The magnetic field of 3 T along the $c$ axis can suppress the resistance drop (green curve in Fig. 3a), which further indicates the emergence of superconductivity at the interface between the PtIr tip and the (001) surface of $TbMn_6Sn_6$. The temperature and magnetic field evolutions of the PCS are presented in Fig. 3b and c, respectively. The PCS at 1.5 K (marked in Fig. 3b) shows clearly two conductance peaks at ±0.6 mV. For a PC, the two conductance peaks are usually taken as the hallmark of Andreev reflection processes at the interface between a normal metal and a superconductor[32]. With the increase of temperature or magnetic field, the two conductance peaks are gradually suppressed, consistent with the signature of weakening superconductivity, confirming the emergence of superconductivity at the $TbMn_6Sn_6$/PtIr tip PC interface.

Interestingly, the MR curve at 1.5 K (Fig. 3d) shows a notable hysteresis loop when applying the magnetic field along the $TbMn_6Sn_6$ $c$ axis ($\mathbf{B} \| c$ axis). Furthermore, this hysteresis loop disappears when the magnetic field is applied along the in-plane direction ($\mathbf{B} \perp c$ axis), as shown in Supplementary Fig. 7. Thus, the MR curves show a similar magnetic anisotropy to the bulk ferrimagnetic $TbMn_6Sn_6$ (Fig. 1c) and $TbMn_6Sn_6$/metallic film heterostructure (Fig. 2), suggesting that the

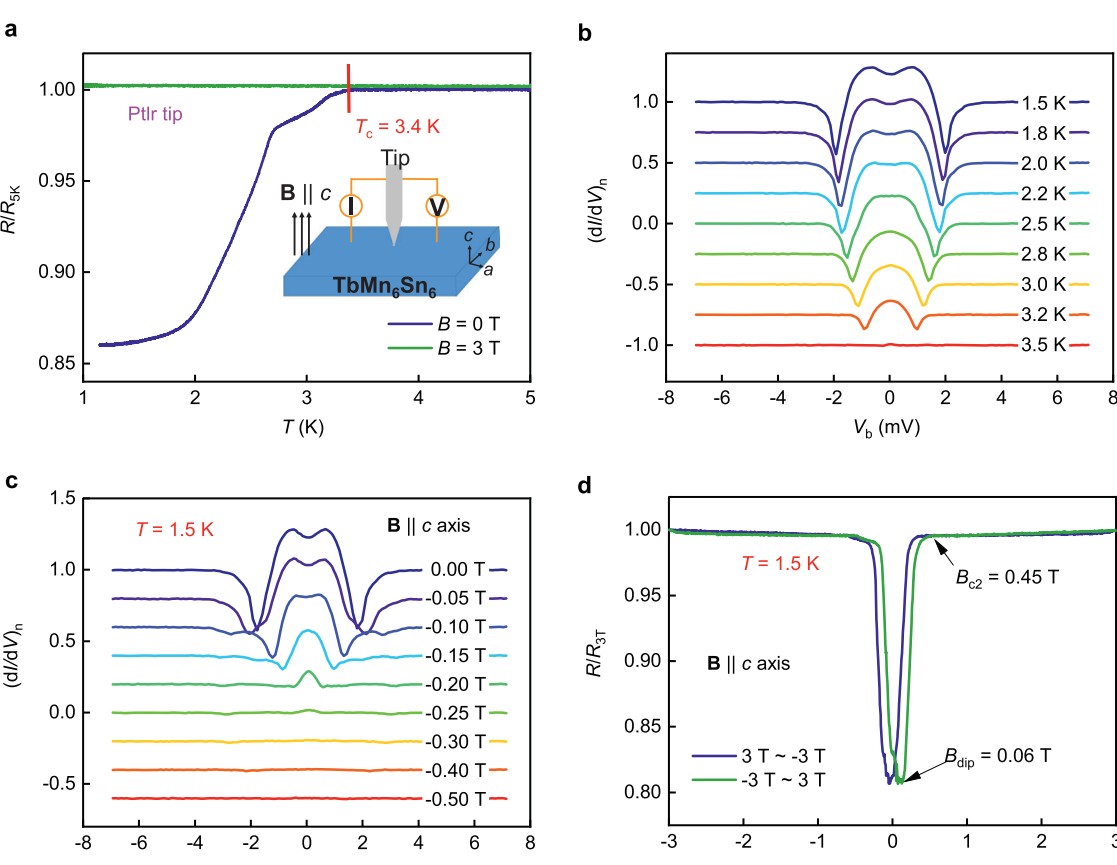

**Fig. 3 | The evidence of superconductivity at the point contact (PC) by pressing the PtIr tip onto the (001) surface of $TbMn_6Sn_6$. a** The temperature dependence of the normalized PC resistance ($R/R_{5K}$) at zero bias with (green curve, $B = 3$ T) or without (blue, $B = 0$ T) applying the magnetic field. Inset: the schematic of the PC configuration, the magnetic field is applied along the out-of-plane direction ($\mathbf{B} \| c$ axis). **b, c** The temperature and magnetic field dependence of the normalized point-contact spectra (PCS). The PCS curves are shifted vertically for clarity. The applied magnetic field is ramping from 0 T to −0.5 T at 1.5 K. **d** The magnetoresistance measurements of the PC at 1.5 K. The PC resistance in the normal state at 5 K is 10.7 Ω.

superconducting state in the PC configuration couples to the magnetization along the out-of-plane direction[33,34] and thus breaks TRS, in agreement with the interface superconductivity in $TbMn_6Sn_6$/metallic film heterostructures discussed above (Fig. 2). We note in passing that the superconductivity can also be induced on the side surface of $TbMn_6Sn_6$ with PtIr tips. The results for one such PC can be found in Supplementary Fig. 8. Similar to the $TbMn_6Sn_6$/Ni film heterostructure (see Supplementary Fig. 2), the superconductivity is also successfully induced by using ferromagnetic Ni tips for PC on $TbMn_6Sn_6$. Related results and discussions can be referred to Supplementary Fig. 9, Fig. 10, and Text I.

To test the reproducibility of the experimental observations presented above, we have probed 92 PC positions and 170 PC states on the different surfaces of two $TbMn_6Sn_6$ samples from the same batch using various metallic tips, including PtIr, Au, Ni, and Ag (see Supplementary Fig. 11a). All these non-superconducting tips are found to induce superconductivity on either the top (001) or side surfaces of $TbMn_6Sn_6$. The statistics of $T_c$ versus the tip materials (see Supplementary Fig. 11b) show no significant variations in the maximum $T_c$ values, whether the tip materials are relatively hard (PtIr tip) or soft (Au tip), paramagnetic or ferromagnetic. These striking observations point to a universal and highly robust interface superconducting state in $TbMn_6Sn_6$/metal heterostructures.

### The structural and elemental analyses of the interface of $TbMn_6Sn_6$/metal heterostructure

To get further insight into this emergent superconducting phase, the structural and elemental mappings have been performed in a high-resolution scanning transmission electron microscopy (STEM) system.

The high-angle annular dark-field STEM (HAADF STEM) image of one typical $TbMn_6Sn_6$ single crystal is shown in Fig. 4a, which manifests a regular atomic stacking sequence along the c-axis of $TbMn_6Sn_6$. Figure 4b shows a STEM image of a heterostructure made by depositing 10 nm Au film on the (001) surface of $TbMn_6Sn_6$, where the interface superconductivity emerges. The top dark region is the carbon shielding layer, which is deposited during the thin lamellae preparation for STEM to prevent Au film from being damaged by the gallium ion beam. The bright stripe region in the middle of the image is the 10 nm thick Au film. The region just below the gold film (roughly marked by the dashed yellow rectangle) does not exhibit long-range ordered crystalline structures, but shows polycrystalline $TbMn_6Sn_6$ structure (Supplementary Text II and Fig. 13). In this work, this region is referred to the degraded $TbMn_6Sn_6$ layer. In the region a little far from the interface, the regular atomic interlayer stacking structure of $TbMn_6Sn_6$ is detected. The element distribution of the $TbMn_6Sn_6$/Au heterostructure can be revealed by the elemental mappings from the energy-dispersive X-ray spectroscopy (EDS). The blue stripe of the Au element distribution shown in Fig. 4c is consistent with the Au film region in Fig. 4b. The Mn and Tb shown in Figs. 4d and e are distributed in both the degraded and crystalline $TbMn_6Sn_6$ region. In Fig. 4f, the Sn element is relatively deficient in the degraded region near the interface compared to the regions with the regular $TbMn_6Sn_6$ atomic lattice, excluding the possibility of superconducting tin filamentary.

To further reveal the origin of the degraded layer, we also conducted STEM studies on as-grown $TbMn_6Sn_6$ samples. The results show that the tin-deficient degraded layer already exists in the as-grown samples, indicating the degraded layer is naturally formed near the surface of $TbMn_6Sn_6$ (Supplementary Text II and Fig. 14).

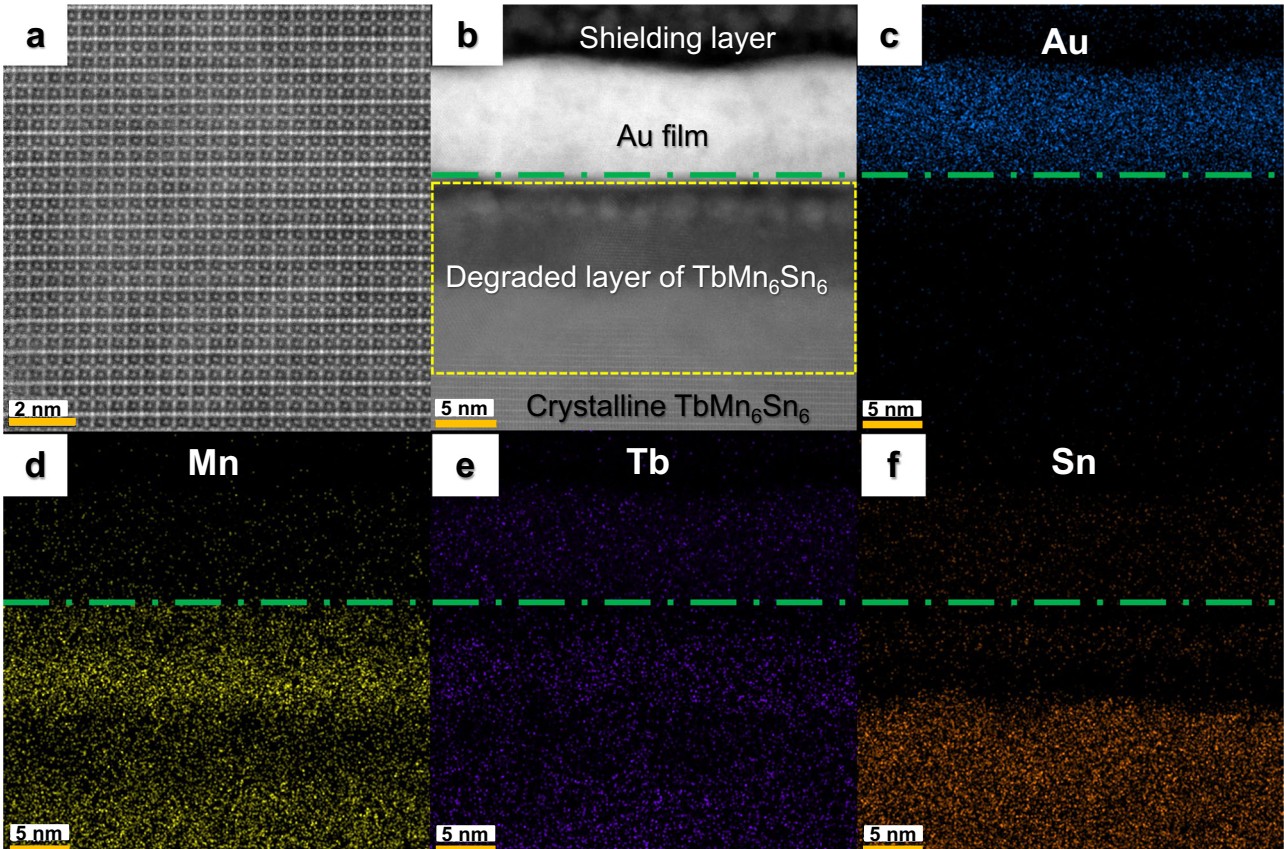

**Fig. 4 | Structural images of the $TbMn_6Sn_6$ single crystal and elemental mappings of the $TbMn_6Sn_6$/Au heterostructure. a, b** The cross-sectional HAADF STEM images of the $TbMn_6Sn_6$ single crystal and the $TbMn_6Sn_6$/Au heterostructure (s8). The heterostructure is fabricated by depositing 10 nm Au film on the (001) surface of the $TbMn_6Sn_6$ single crystal. **c-f** The energy-dispersive X-ray spectroscopy (EDS) mappings of Au, Mn, Tb, and Sn elements distribution of the structural region shown in (**b**). The interface of the $TbMn_6Sn_6$/Au heterostructure is marked by a green dashed line.

## Discussion

In summary, we discovered emergent superconductivity at the interface of TbMn$_6$Sn$_6$/metal heterostructures formed by depositing the non-superconducting metallic thin films, such as the nonmagnetic Au and Ag, and ferromagnetic Ni, on the surfaces of ferrimagnetic TbMn$_6$Sn$_6$. The superconductivity is also evidenced at PC interfaces between TbMn$_6$Sn$_6$ and non-superconducting metallic tips, including ferromagnetic Ni tip. We found that the emergent superconductivity at the interface is quasi-2D and couples to the magnetization inherited from the magnetic order of TbMn$_6$Sn$_6$ and exhibits hysteretic magnetoresistance. These findings suggest the emergence of a quasi-2D time-reversal-symmetry-breaking superconducting state at the interface of TbMn$_6$Sn$_6$/metal heterostructures.

A plausible mechanism is that the carrier doping may take place near the interface and enable an emergent quasi-2D superconductor. Furthermore, near the interface, polycrystalline TbMn$_6$Sn$_6$ structures are revealed in the degraded layer by structural analyses (Supplementary Text II and Fig. 13). Consequently, the magnetization and strong spin-orbit coupling (SOC) from polycrystalline TbMn$_6$Sn$_6$ are expected to be inherited by the emergent interface superconductivity. As a result, the interface of the TbMn$_6$Sn$_6$/metal heterostructure contains, surprisingly, all the essential ingredients to generate chiral topological superconductivity, as proposed for 2D Rashba SOC semiconductors proximity coupled to an $s$-wave superconductor and a ferromagnetic insulator[26,27], as illustrated in Fig. 5a. However, the theoretical proposal has not been realized presumably because the difficulty to meet the condition that proximity-induced superconductivity and ferromagnetism are coupled to each other in the semiconductor layer. These challenges are overcome in the TbMn$_6$Sn$_6$/metal heterostructures, since the quasi-2D superconductivity coupled with the magnetization and strong SOC are all established at the interface (Fig. 5b), thus providing a more realistic material platform for realizing chiral topological superconductivity. More specifically, the Rashba SOC leads to a pair of spin nondegenerate bands in the $k_x$-$k_y$ plane (Fig. 5c). The exchange field from the degraded TbMn$_6$Sn$_6$ layer opens a Zeeman energy gap ($E_0$) at the crossing point of these two bands at $k_x = k_y = 0$ as shown in Fig. 5d. The emergent interface superconductivity opens up a superconducting gap $\Delta$ in the outer band if the chemical potential $\mu$ is located in the gap $E_0$. When the condition $E_0 > (\mu^2 + \Delta^2)^{1/2}$ is satisfied, the proposed effective chiral $p$-wave superconducting

state[26,27] can be realized at the interface of the TbMn$_6$Sn$_6$/metal heterostructures. We note in passing that although the direct evidence has not been found at the current stage, if the interface superconductivity takes place in the kagome layers, topological superconductivity can also arise, which is discussed using a theoretical model in the Supplementary Text III and IV for different effective exchange couplings. Our findings provide a new direction for studying the interplay of ferromagnetism and superconductivity at the interface of kagome magnet and metal heterostructures, and at the same time, uncover a material platform to pursue time-reversal symmetry-breaking topological superconductivity.

## Methods

### Electrical transport and magnetization measurements
The electrical transport measurements of longitudinal and Hall resistance were performed in a commercial physical property measurement system (PPMS, Quantum Design) by using the standard four-electrode method. Vibrating sample magnetometer (VSM) magnetization measurements were carried out in a commercial magnetic property measurement system (MPMS3, Quantum Design).

### Cryogenic point contact measurements
The cryogenic point contact system contains the low-temperature platform and the PC system. The point contact is performed on the top stage of three-axis piezo stacks from Attocube by pressing sharp metallic tips onto the TbMn$_6$Sn$_6$ sample's selected surface. The point-contact spectra are measured by using the standard lock-in method. The cryogenic circumstance is created by inserting the piezo stacks into the Leiden dilution refrigerator (CF450) with three-axis superconducting magnets (1 T/1 T/3 T). The lowest temperature used for the point contact experiments is 1.1 K in this work. More discussions of the point-contact Andreev-reflection spectroscopy can be found in Supplementary Text V in the Supplementary Information.

### Single-crystal growth
High-quality single crystals of TbMn$_6$Sn$_6$ were prepared by using a self-tin-flux growth method. Terbium ingots, manganese pieces, and tin lumps with a molar ratio of 1:6:20 were packed into an alumina crucible blocked with a piece of quartz wool on the top, which was then sealed in a fused quartz ampoule under vacuum. The ampoule was heated to

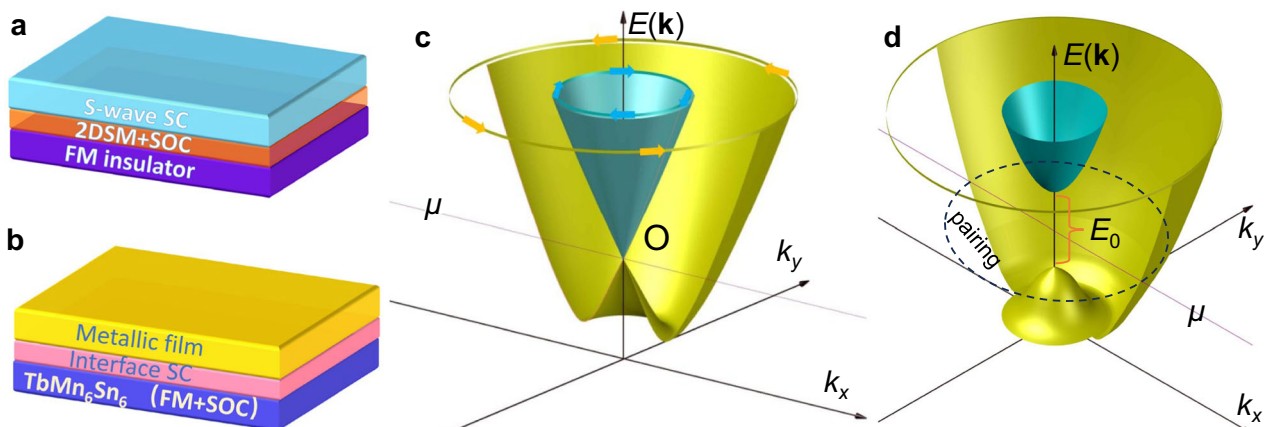

**Fig. 5 | The theoretical scenario for the interface superconductivity of TbMn$_6$Sn$_6$/metal heterostructure. a** The strategy of the proximity-induced topological superconductivity in stacked layers formed by an $s$-wave superconductor (SC), a 2D semiconductor (2DSM) with Rashba spin-orbit coupling (SOC), and a ferromagnetic (FM) insulator. **b** The schematic of the detected interface superconductivity in TbMn$_6$Sn$_6$/ metal heterostructures. **c** The schematic of the electron band structure $E(\mathbf{k})$ of the 2D degraded layer with Rashba SOC without considering the exchange field from TbMn$_6$Sn$_6$. The opposite spin

orientations along the effective Rashba SOC field direction are denoted by yellow and cyan arrows. The band with a yellow (cyan) color indicates this band possesses the same spin texture denoted by the yellow (cyan) arrows. Chemical potential $\mu$ is marked by a purple line. **d** The schematic of energy dispersion of the 2D degraded layer with Rashba SOC coupled to the kagome magnet TbMn$_6$Sn$_6$. $E_0$ is the Zeeman gap. The dashed circle marked by 'pairing' indicates the Cooper pairing in the outer band when the chemical potential $\mu$ is located in the gap $E_0$.

1000 °C and kept for a few hours, then slowly cooled down to the centrifuging temperature of 600 °C. While hot, the ampoule was inverted and spun in a centrifuge, forcing the excess tin into the quartz wool located in the crucible. This method produced several hexagonal flat single crystals of millimeters in size. To ensure a clean surface, some samples are mechanically polished before measurements. For some large crystals with a size up to 2 mm × 2 mm × 0.5 mm, the fresh (001) surfaces are obtained by a mechanically cleaving process. All the clean side surfaces are obtained by the polishing process.

## Scanning transmission electron microscopy (STEM) measurements

All the STEM samples were prepared by the focused ion beam (FIB, Helios G4) system. All samples were protected by depositing a 10 nm carbon layer in the LEICAEM ACE200 coating system before transferring them to the FIB system. Before the FIB treatment, a 2 μm carbon protection layer was further deposited under 30 kV, 0.1 nA experimental condition in the FIB system. Then, the samples were thinned using an accelerating voltage of 30 kV with a decreasing current from 240 to 50 pA, followed by a fine polish with an accelerating voltage of 2 kV with a current of 20 pA to reduce surface damage. A Titan Cubed Themis G2 double Cs-corrected scanning transmission electron microscope was used at 300 kV to obtain the HAADF STEM images and energy-dispersive X-ray spectroscopy (EDS) mappings with a probe-forming semi-angle of 30 mrad. The HAADF STEM images were acquired with a beam current of ~50 pA and a collection semi-angle snap in the range of 39–200 mrad, 145 mm camera length.

## Sample treatment and film deposition

Before preparing the $TbMn_6Sn_6$/metallic film heterostructures, pre-treating processes were applied to the $TbMn_6Sn_6$ samples, including mechanically cleaving (Fig. 2a, b), surface polishing (Fig. 2c, d, Supplementary Figs. 2 and 5), or only ultrasonic cleaning by ethanol and acetone (Fig. 2e–f, Supplementary Figs. 4, 15 and 20). The $TbMn_6Sn_6$ samples used in $TbMn_6Sn_6$/Au heterostructure s7 and s8 are obtained by mechanically cleaving one $TbMn_6Sn_6$ crystal into two parts. Subsequently, the samples were transferred to the e-beam eva-poration chamber of an LJUHV E-400L E-Beam Evaporator. The 10 nm-thick Au (or 8 nm-thick Ni) films were deposited onto the sample surface at room temperature by the standard e-beam evaporation method with the electron energy of 8 keV. The chamber pressure is lower than $2 \times 10^{-6}$ Torr before deposition and was maintained below $4 \times 10^{-6}$ Torr during the evaporation process. The purities of the commercial Au and Ni targets are 99.999%, and a deposition rate of 0.5 Å/s (0.3 Å/s) was achieved by focusing the electron beam on the Au (Ni) target. The fabrication processes of heterostructures s7 and s8 are identical. The 10-nm thick Ag film in Supplementary Fig. 6 was grown at room temperature by using the direct current (DC) reactive magnetron sputtering method with the MSP-3200 Magnetron Sputtering System. In the sputtering chamber, the Ag film (target impurity, 99.99%) was deposited onto the $TbMn_6Sn_6$ surface at a rate of 2.7 Å/s with an Ar gas pressure of 0.9 Pa.

## Data availability

All data needed to evaluate the conclusions in the study are present in the paper and/or the Supplementary Information. The data that support the findings of this study are available from the corresponding authors upon request.

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

## Acknowledgements

We thank Ying Xing, Yi Liu, Jingmin Zhang, Jun Xu, Xuetao Di, and Pengfei Zhan for their help with the experiments. This work was financially supported by Beijing Natural Science Foundation [Z180010 (J.W.; H.W.)], National Key Research and Development Program of China [2018YFA0305600 (J.W.; S.J.), 2019YFA0308403 (H. Jiang)], National Natural Science Foundation of China [11888101 (J.W.)], the Innovation Program for Quantum Science and Technology [2021ZD0302403 (J.W.)], China Postdoctoral Science Foundation [2021M700253 (Y.L.)], U.S. Department of Energy, Basic Energy Sciences [DE-FG02-99ER45747 (Z.W.)], the Cottrell SEED Award from Research Corporation for Science Advancement [27856 (Z.W.)], Beijing Natural Science Foundation [1202005 (H.W.)], Capacity Building for Sci-Tech Innovation-Fundamental Scientific Research Funds [20530290057 (H.W.), KM202010028014 (H.W.)], Academy for Multidisciplinary Studies, Capital Normal University (H.W.).

## Author contributions

J.W. conceived and instructed the experiments. H.W. conducted cryogenic point contact experiments. Y.L. performed the transport measurements. M.G., H. Jiang, Z.W., and X.C.X. developed the theoretical models. X.G. and P.G. performed the TEM measurements. W.M. and S.J. grew the samples. H.W., Y.L., J.L., H. Ji, and J.G. fabricated the heterostructures. H.W., Y.L., and J.W. wrote the manuscript with theoretical input from M.G., H. Jiang, and Z.W.

## Competing interests

The authors declare no competing interests.
