## [Peer Review File · Nature Communications]

Emergent superconductivity in topological-kagome magnet/
metal heterostructuresReviewers' Comments:

Reviewer #1:

Remarks to the Author:

The authors report on interfacial superconductivity at the interface between the Kagome magnet TbMn_6Sn_6 and a metal. This is a very interesting result because it may be the long-term-awaited time-reversal symmetry-breaking spin-triplet superconductivity. The paper is highly appreciated by the condensed matter community. Hence, I recommend the paper may be considered for publication. However, there are some concerns, which the authors need to clarify.

The main concern is that the authors state that the degraded layer may be responsible for the interfacial superconductivity in metal/ TbMn_6Sn_6 . Based on the cross-sectional TEM observations, the authors stated that the degradation layer at the Au/ TbMn_6Sn_6 interface (in the region of about 18 nm from Au) is due to stress generated during surface polishing and gold deposition. If this area does not have the crystal structure of TbMn_6Sn_6 , then all discussions of superconductivity at the Au/ TbMn_6Sn_6 interface are meaningless. In the latter part of the article, the author states that "the degraded TbMn_6Sn_6 layer has a different carrier concentration and weaker magnetism, which may make the appearance of superconductivity more realistic," but if it is not a crystalline layer, then this argument is also meaningless. By the way, this degraded layer is significantly deficient in Sn. The observation is Au/ TbMn_6Sn_6 , but the discussion is also substituted with metal/ TbMn_6Sn_6 in the middle of the article. Does this mean that the same thing can happen with Ag/ TbMn_6Sn_6 or Ni/ TbMn_6Sn_6 ?

How do you explain the superconductivity observed in PCS? In this case, the stress caused by surface polishing or metal film deposition does not seem to have any effect. TEM observation of the as-grown sample would be necessary to eliminate the influence of the metal film deposition. Observations on as-grown samples may be necessary to confirm this point.

Some minor point: the superconductivity observed in PCS seems to undergo a two-step transition from Fig. 3a, but there is no explanation for this. The vertical axis is normalized by the resistance at 5 K, not 4 K. The same is true for Fig. 3d.

Reviewer #2:

Remarks to the Author:

Due to the novel properties and the potential application for topological fault-tolerant quantum computation, the topological superconductor, which was predicted to host Majorana fermion excitations, has attracted great attention in condensed matter physics. In this article by H. Wang et al., the authors report the emergent superconductivity at the interface of kagome Chern magnet TbMn_6Sn_6 and metal heterostructures. The theme of their study is indeed timely, and the three results revealed in the paper are very important and interesting. i) The TbMn_6Sn_6 single crystals show ferrimagnetic behavior below the Curie temperature ≈ 423 K. It is an impressive discovery that superconductivity can emerge in TbMn_6Sn_6 single crystals with such high Curie temperature. ii) The observed superconductivity could be topologically nontrivial, because it contains all the essential ingredients for generating chiral topological superconductivity in a two-dimensional (2D) system, including strong spin-orbit-coupling, the quasi-2D superconductivity coupled to the ferromagnetism. iii) The emergent superconductivity at the heterostructure interface between two different conductors represents a new class of interface superconductivity, which may shed new light on the underlying mechanism of interface superconductivity and unconventional superconductivity.

In my opinion, this is a high-quality research paper for exploring new superconductors and chiral topological superconductivity via the interface engineering of the heterostructure composed of magnetic topological materials and non-superconducting metals. A broad audience could be inspired by this work. Thus, I would like to suggest the publication of the manuscript after

addressing the following comments.

- i. For the R-T curves of the TbMn₆Sn₆/Au heterostructure shown in Fig. 2, the resistance drops are suppressed by magnetic fields, suggesting the observation of superconductivity. At low temperatures, the detected interface superconductivity doesn't show zero resistance. Do the authors have any possible explanations for the residual resistance?
- ii. As shown in Fig. 2c and Fig. 3d, when the superconductivity is emergent, the coercive field of the TbMn₆Sn₆/metal heterostructure is much smaller than that of bulk TbMn₆Sn₆ (1.8 T at 2 K, Fig. 1c). It is better to add the discussion on this difference in the manuscript.
- iii. As the potential readers for this paper may not be experts on point-contact Andreev spectroscopy, the additional information inserted into the supplementary file would be helpful.
- iv. Some typos should be avoided. Such as, 'because' in Line 293, page 10, should be 'because of'; 'magnetos' in Line 350, page 12, should be 'magnets'.

Reviewer #3:

Remarks to the Author:

The authors report the emergence of superconductivity at the interface between metals and the kagome magnet TbMn₆Sn₆. The interfaces are created through metal thin film coating or metal-tip point contact on single crystal TbMn₆Sn₆ samples, and the emergence of superconductivity is suggested by the following observations: 1) a lowering of resistance and its suppression under external magnetic fields in transport measurements for metal thin films/TbMn₆Sn₆, and 2) an enhancement of conductance in point-contact conductance spectra for single TbMn₆Sn₆. The authors point out that the emergent superconductivity could be unconventional; however, in my opinion, their experimental data are NOT enough and solid to support their idea. In this sense, the paper is NOT suitable for publication in Nature Communication. The issues have been listed as follows:

- 1) Since zero-resistance has not been observed in the transport measurement, one surely can suspect the formation of superconducting filamentary or impurities in TbMn₆Sn₆ (or in the degraded layer) by the thin film coating and point contact (i.e., due to mechanical force). Based on this suspicion, many of their interpretations are not compelling. For example, from the hysteretic magnetoresistance (MR) behavior, the authors suggest the superconductivity coupled with the magnetization of TbMn₆Sn₆, but it could be independent of each other. The magnetic layer is also still underneath.
- 2) The authors suggest the degraded layer is superconducting, but there is no reason to have inherent magnetization and strong SOC from TbMn₆Sn₆ because it is no longer to be TbMn₆Sn₆. They need to secure specific information for the degraded layer, like STM or ARPES measurement, to suggest unconventional superconductivity. Isn't it possible to remove the metal thin film atop and perform other measurements?
- 3) More systematic PCS measurements need to be carried out, and more detailed information on their measurements should be provided. According to their interpretation, the mechanical force to make a point contact is expected to lead to a local degradation and superconductivity on the surface of TbMn₆Sn₆, and the degradation would be worse with increasing the applied force, which could be estimated from contact resistance. The contact resistance is not directly indicative of the diameter of a point contact — actual contact can consist of many parallel contacts due to the microstructures of a tip and sample.
- 4) Along with comment 3), the authors need to clarify the origin of the emergent superconductivity, which is acceptable in parallel for both systems: metal thin film/TbMn₆Sn₆ and point-contact/TbMn₆Sn₆. I don't think the point contact does make such a huge degradation (e.g., stoichiometry change) as observed in the TEM and EDX measurements on the Au/TbMn₆Sn₆.
- 5) In the conductance spectra, significant dips are consistently observed, which may narrate the thermal regime or so-called Maxwell component of the point contact resistance. In such a regime, the conductance spectra may reflect temperature dependence resistance, not only spectroscopic information. The observation of the zero-bias conductance peak-like feature (in SI) could also be

due to the same reason. They need to revisit the contact resistances and provide information on those.

6)The authors need to be more careful in preparing the figures in the manuscript and SI. For example:

The tick direction is not consistent (Figure 2 and Figure 3).

The scale bars are too small and not visible (Figure 4)

No x-axis (top and bottom) (Supplementary Figure 2, Figure 8, Figure 12 (a) and (b), probably crop error)

No label for supplementary Fig. 12 (a, b, d, e).

7)The authors need to add more information on the fabrication of the metal thin films, like a method, deposition temperature, Etc. They may find a deposition condition that would not lead to surface degradation.

In the following, we addressed the individual questions. The reviewers' original comments are shown by blue characters. Our responses are shown by black characters.

Reviewer comments

Reviewer #1 (Remarks to the Author):

Comment:

The authors report on interfacial superconductivity at the interface between the Kagome magnet TbMn_6Sn_6 and a metal. This is a very interesting result because it may be the long-term-awaited time-reversal symmetry-breaking spin-triplet superconductivity. The paper is highly appreciated by the condensed matter community. Hence, I recommend the paper may be considered for publication. However, there are some concerns, which the authors need to clarify.

Response: We sincerely thank Reviewer #1 for his/her very high assessments and recommendation of publication of our manuscript. Reviewer #1 indeed perfectly captures the essential significance of our manuscript and the great importance of the emergent superconductivity in topological kagome magnet/metal heterostructures. In the following, we provide detailed discussions and new experimental results on as-grown TbMn_6Sn_6 and $\text{Au/TbMn}_6\text{Sn}_6$ heterostructures according to the reviewer's constructive comments and suggestions, which we hope will help to clarify the reviewer's concern.

Comment:

The main concern is that the authors state that the degraded layer may be responsible for the interfacial superconductivity in metal/ TbMn_6Sn_6 . Based on the cross-sectional TEM observations, the authors stated that the degradation layer at the $\text{Au/TbMn}_6\text{Sn}_6$ interface (in the region of about 18 nm from Au) is due to stress generated during surface polishing and gold deposition. If this area does not have the crystal structure of TbMn_6Sn_6 , then all discussions of superconductivity at the $\text{Au/TbMn}_6\text{Sn}_6$ interface are meaningless. In the latter part of the article, the author states that "the degraded TbMn_6Sn_6 layer has a different carrier concentration and weaker magnetism, which may make the appearance of superconductivity more realistic," but if it is not a crystalline layer, then this argument is also meaningless. By the way, this degraded layer is significantly deficient in Sn. The observation is $\text{Au/TbMn}_6\text{Sn}_6$, but the discussion is also substituted with metal/ TbMn_6Sn_6 in the middle of the article. Does this mean that the same thing can happen with $\text{Ag/TbMn}_6\text{Sn}_6$ or $\text{Ni/TbMn}_6\text{Sn}_6$?

Response: We greatly appreciate the valuable feedback provided by Reviewer #1, which inspires us to conduct further detailed research on the degraded TbMn_6Sn_6 layer. We further analyzed the lattice structures in the degraded layer of the $\text{TbMn}_6\text{Sn}_6/\text{Au}$

heterostructure. In the degraded layers of two typical TbMn₆Sn₆/Au heterostructures s8 (shown in Fig. 4 in the main text) and s17 (more transport, TEM, and EDS results of s17 can be found in Fig. R3 and Fig. R4), we can identify some regions showing lattice structures that are consistent with the crystal structure of TbMn₆Sn₆. Fig. R1 displays three high-angle annular dark-field STEM (HAADF STEM) images of the degraded layers in s8 (Fig. R1a) and s17 (R1b-c). These images present regions showing crystal structures in the degraded layer, and the interplanar lattice spacing values in these regions are approximately 4.5 Å. Using the PDF Card (NO. 01-072-3490) as a reference, this value corresponds to the space distance of (002) TbMn₆Sn₆ along the [001] crystal orientation (4.502 Å). These findings suggest that the degraded layer mainly consists of the polycrystalline TbMn₆Sn₆. We hope that the new TEM results can remove Reviewer #1's concerns about the crystallinity of TbMn₆Sn₆ in the degraded layer.

Fig. R1 | Further analyses of the microstructure of the degraded TbMn₆Sn₆ layer near the interface of two typical TbMn₆Sn₆/Au heterostructures s8 (a) and s17 (b-c). The TbMn₆Sn₆/Au heterostructures are fabricated by depositing 10 nm Au film on the (001) surface of the TbMn₆Sn₆ single crystal. The selected area is marked by white squared frames. **a-c** Cross-sectional HAADF STEM images of the crystal lattice in the degraded TbMn₆Sn₆ layer near the interface of TbMn₆Sn₆/Au heterostructure s8 (a) and s17 (b-c). The interplanar spacing values of the lattices are marked by two red lines. The white arrows indicate the [001] orientation of the bulk TbMn₆Sn₆ single crystal.

To further understand the nature of degraded layers, we also characterize the as-grown TbMn₆Sn₆ samples for comparison, which are shown in Fig. R2. Similar to the TEM data of Au/TbMn₆Sn₆ heterostructure shown in Fig. 4 in the manuscript, the TEM image of the as-grown TbMn₆Sn₆ also shows that the degraded layer exists near the sample surface. Furthermore, similar to the Au/TbMn₆Sn₆ heterostructure, the deficiency of the Sn element is also observed in the degraded region near the surface of the as-grown TbMn₆Sn₆ (Fig. R2d). Therefore, the degraded layer already exists before the metal electrode (Au, Ag, and Ni) deposition. Based on these new experimental results, we

have changed the statement in our revised manuscript on Page 8 Line 276 to “The results show that the tin-deficient degraded layer already exists in the as-grown samples, indicating the degraded layer is naturally formed near the surface of TbMn₆Sn₆.”

Fig. R2 | Structural image and elemental mappings of the as-grown TbMn₆Sn₆ single crystal. **a** A cross-sectional HAADF STEM image of the TbMn₆Sn₆ single crystal surface. **b-d** The EDS mappings of Tb, Mn, and Sn elements distribution of the HAADF STEM image **a**. The interface between the TbMn₆Sn₆ and the shielding layer, a carbon protecting layer with the thickness of more than 100 nm, is marked by a green dashed line. The degraded TbMn₆Sn₆ layer is marked by a yellow dashed line box.

We also fabricated a new TbMn₆Sn₆/Au heterostructure (s17), wherein the TbMn₆Sn₆ is only ultrasonic cleaned by ethanol and acetone to avoid the possible influence of surface polishing. The emergent superconductivity of s17 is detected by electronic transport measurements, as shown in Fig. R3. A notable resistance drops down to 54% of R_{6K} is observed, and the superconducting transition temperature is about 3.8 K,

consistent with the results of other heterostructures shown in our manuscript.

Fig. R3 | Temperature dependence of the normalized longitudinal resistance of the TbMn₆Sn₆/Au heterostructure (s17).

After confirming the superconductivity of the heterostructure s17, we conduct TEM and EDS measurements (Fig. R4) on the interface of TbMn₆Sn₆/Au film, which show a similar degraded layer and tin-deficient elemental mapping as the TbMn₆Sn₆/Au heterostructure shown in Fig. 4 in the manuscript and as-grown TbMn₆Sn₆ sample (Fig. R2). Considering that degraded layers with consistent elemental mapping are observed near the surface of all samples measured by TEM, especially near the surface of as-grown TbMn₆Sn₆, it is reasonable to conclude that such degraded layers exist near the interface of all metal/TbMn₆Sn₆ heterostructures, including Ag/TbMn₆Sn₆ or Ni/TbMn₆Sn₆. In the revised manuscript, we have added further experimental results to reveal the properties of the degraded layer.

Fig.R4 | Structural image and elemental mappings of TbMn₆Sn₆/Au heterostructure (s17). **a-b** The cross-sectional HAADF STEM images of the TbMn₆Sn₆ single crystal and the TbMn₆Sn₆/Au heterostructure. The heterostructure is fabricated by depositing 10 nm Au film on the (001) surface of the TbMn₆Sn₆ single crystal. **c-f** The EDS mappings of Au, Tb, Mn, and Sn elements in the HAADF STEM image **b**. The interface between the TbMn₆Sn₆ and the Au film is marked by a green dashed line. The degraded TbMn₆Sn₆ layer is marked by a yellow dashed line box.

Comment:

How do you explain the superconductivity observed in PCS? In this case, the stress caused by surface polishing or metal film deposition does not seem to have any effect. TEM observation of the as-grown sample would be necessary to eliminate the influence of the metal film deposition. Observations on as-grown samples may be necessary to confirm this point.

Response: We thank Reviewer #1 for this valuable suggestion. As mentioned above, we have followed Reviewer #1's advice and performed the TEM and EDS measurements on the as-grown TbMn₆Sn₆ sample (Fig. R2). As shown in Fig. R2, the degraded layer is observed near the surface of the as-grown TbMn₆Sn₆, which demonstrates that the degraded layer is not caused by surface polishing or metal film deposition. Therefore, when a metallic tip contacts the surface of TbMn₆Sn₆, the situation is similar to the metal film/ TbMn₆Sn₆.

In the revised version of the manuscript, we have added related discussions in the main text and supplementary Information.

Comment:

Some minor point: the superconductivity observed in PCS seems to undergo a two-step transition from Fig. 3a, but there is no explanation for this. The vertical axis is normalized by the resistance at 5 K, not 4 K. The same is true for Fig. 3d.

Response: The two-step transition shown in the R - T curve usually implies the existence of two superconducting phases in the sample. In our work, the interface conditions between the metallic tips and the surface of TbMn_6Sn_6 have a notable influence on the emergence of superconductivity. When the PtIr tip contacts the surface of TbMn_6Sn_6 , the complex interfacial conditions in the contact region may trigger different superconducting phases and the multiple-step (including two-step) transition in the R - T curves.

We also thank the reviewer for the reminder. The normal state resistances shown in Fig. 3a at 4 K and 5 K are almost the same. In the revised manuscript, we normalize the vertical axis by the resistance at 5 K as suggested.

Reviewer #2 (Remarks to the Author):

Comment:

Due to the novel properties and the potential application for topological fault-tolerant quantum computation, the topological superconductor, which was predicted to host Majorana fermion excitations, has attracted great attention in condensed matter physics. In this article by H. Wang et al., the authors report the emergent superconductivity at the interface of kagome Chern magnet TbMn_6Sn_6 and metal heterostructures. The theme of their study is indeed timely, and the three results revealed in the paper are very important and interesting. i) The TbMn_6Sn_6 single crystals show ferrimagnetic behavior below the Curie temperature ≈ 423 K. It is an impressive discovery that superconductivity can emerge in TbMn_6Sn_6 single crystals with such high Curie temperature. ii) The observed superconductivity could be topologically nontrivial, because it contains all the essential ingredients for generating chiral topological superconductivity in a two-dimensional (2D) system, including strong spin-orbit-coupling, the quasi-2D superconductivity coupled to the ferromagnetism. iii) The emergent superconductivity at the heterostructure interface between two different conductors represents a new class of interface superconductivity, which may shed new light on the underlying mechanism of interface superconductivity and unconventional superconductivity.

In my opinion, this is a high-quality research paper for exploring new superconductors and chiral topological superconductivity via the interface engineering of the heterostructure composed of magnetic topological materials and non-superconducting metals. A broad audience could be inspired by this work. Thus, I would like to suggest the publication of the manuscript after addressing the following comments.

Response: We greatly thank Reviewer #2 for his/her recommendation for the publication of our work. We are also grateful for Reviewer #2's high evaluation of our work by saying that "this is a high-quality research paper for exploring new superconductors and chiral topological superconductivity via the interface engineering of the heterostructure composed of magnetic topological materials and non-superconducting metals". Below we give our point-by-point response to Reviewer #2's detailed comments, which are very helpful to improve our manuscript.

Comment:

i. For the R-T curves of the $\text{TbMn}_6\text{Sn}_6/\text{Au}$ heterostructure shown in Fig. 2, the resistance drops are suppressed by magnetic fields, suggesting the observation of superconductivity. At low temperatures, the detected interface superconductivity doesn't show zero resistance. Do the authors have any possible explanations for the residual resistance?

Response: We thank the reviewer for supporting our observation of the emergent superconductivity in the $\text{TbMn}_6\text{Sn}_6/\text{Au}$ heterostructure. Non-zero residual resistance in

the superconducting state, which can be caused by various mechanisms, is not a rare phenomenon in low-dimensional or interface superconductors [Nature 572, 215-219 (2019); Nature Communications 7, 11210 (2016), etc.]. In our work, the residual resistance may result from discontinuous superconducting areas at the interface. As illustrated in the manuscript, the larger resistance drop in the TbMn₆Sn₆/silver film than the TbMn₆Sn₆/silver epoxy indicates better contact between the metallic film and the sample surface is conducive to the formation of superconductivity. Due to the non-van der Waals nature of TbMn₆Sn₆ crystals, it is challenging to obtain a large area of atomic-level flat surfaces from the micrometer-scale TbMn₆Sn₆ single crystals. As a result, when depositing a metallic film onto the TbMn₆Sn₆ surface, it is difficult to achieve a uniform and consistent contact condition, which may lead to inhomogeneous superconductivity with non-zero residual resistance at low temperatures.

In the revised version of the manuscript, we have updated related information in the main text.

Comment:

ii. As shown in Fig. 2c and Fig. 3d, when the superconductivity is emergent, the coercive field of the TbMn₆Sn₆/metal heterostructure is much smaller than that of bulk TbMn₆Sn₆ (1.8 T at 2 K, Fig. 1c). It is better to add the discussion on this difference in the manuscript.

Response: We thank Reviewer #2 for this comment. The discrepancy between the coercive field of the superconducting TbMn₆Sn₆/metal heterostructure and bulk TbMn₆Sn₆ may come from the degraded layer near the interface. As discussed in our manuscript, the properties of interface superconductivity are expected to be influenced by the degraded layer near the TbMn₆Sn₆/metal interface. We further analyzed the lattice structures in the degraded layer of the TbMn₆Sn₆/Au heterostructure. In the degraded layers of two typical TbMn₆Sn₆/Au heterostructures s8 (shown in Fig. 4 in the main text) and s17, we can identify some regions showing lattice structures that are consistent with the crystal structure of TbMn₆Sn₆. Fig. R1 displays three high-angle annular dark-field STEM (HAADF STEM) images of the degraded layers in s8 (Fig. R1a) and s17 (R1b-c). These images present regions showing crystal structures in the degraded layer, and the interplanar lattice spacing values in these regions are approximately 4.5 Å. Using the PDF Card (NO. 01-072-3490) as a reference, this value corresponds to the space distance of (002) TbMn₆Sn₆ along the [001] crystal orientation. These findings suggest that the degraded layer mainly consists of the polycrystalline

TbMn₆Sn₆. Thus, the magnetism in the degraded layer may be weakened by the grain boundary and disorders, and show a smaller coercive field than bulk TbMn₆Sn₆. Therefore, the observed coercive field of the TbMn₆Sn₆/metal heterostructure is much smaller than that of bulk TbMn₆Sn₆.

Fig. R1 | Further analyses of the microstructure of the degraded TbMn₆Sn₆ layer near the interface of two typical TbMn₆Sn₆/Au heterostructures s8 (a) and s17 (b-c). The TbMn₆Sn₆/Au heterostructures are fabricated by depositing 10 nm Au film on the (001) surface of the TbMn₆Sn₆ single crystal. The selected area is marked by white squared frames. **a-c** Cross-sectional HAADF STEM images of the crystal lattice in the degraded TbMn₆Sn₆ layer near the interface of TbMn₆Sn₆/Au heterostructure s8 (**a**) and s17 (**b-c**). The interplanar spacing values of the lattices are marked by two red lines. The white arrows indicate the [001] orientation of the bulk TbMn₆Sn₆ single crystal.

In the revised version of the manuscript, we have updated related information in the main text and Supplementary Information.

Comment:

iii. As the potential readers for this paper may not be experts on point-contact Andreev spectroscopy, the additional information inserted into the supplementary file would be helpful.

Response:

We thank the reviewer for the nice advice. We have added more discussions of the point-contact Andreev-reflection spectroscopy as Supplementary Text V in the Supplementary Information. Details are as follows:

The point-contact Andreev-reflection spectroscopy is a powerful tool for studying the properties of superconducting materials [J. Phys.: Condens. Matter. 10, 8905 (1998); Rep. Prog. Phys. 79, 094502 (2016)]. By employing the Andreev reflection process [Sov. Phys. JETP 19, 1228 (1964)], which occurs when an electron in the normal metal is reflected as a hole at the interface between the superconductor and normal metal, the

point-contact Andreev-reflection spectroscopy can be used to measure the superconducting gap, determine the superconducting order parameter, analyze the pairing symmetry, and understand the pairing mechanism [Rev. Mod. Phys. 77, 109 (2005); Supercond. Sci. Technol. 23, 043001 (2010); Sci. Bull. 63, 1141 (2018)]. The point contact with the size much smaller than the electron elastic mean free path, locates in the ballistic regime. For the size larger than the elastic mean free path, but smaller than the inelastic mean free path, the point contact locates in the intermediate regime [J. Phys. C: Solid State Phys. 13 6073 (1980)]. When point-contact Andreev-reflection measurements are conducted on a conventional superconductor, there would be only double conductance peaks in the PCS for the case in the ballistic regime, and double conductance peaks combined with double conductance dips in the PCS for the case in the intermediate regime.

Comment:

iv. Some typos should be avoided. Such as, ‘because’ in Line 293, page 10, should be ‘because of’; ‘magnetos’ in Line 350, page 12, should be ‘magnets’.

Response:

We thank the reviewer for the kind reminders. We have corrected these typos in the revised manuscript.

Reviewer #3 (Remarks to the Author):

Comment:

The authors report the emergence of superconductivity at the interface between metals and the kagome magnet TbMn_6Sn_6 . The interfaces are created through metal thin film coating or metal-tip point contact on single crystal TbMn_6Sn_6 samples, and the emergence of superconductivity is suggested by the following observations: 1) a lowering of resistance and its suppression under external magnetic fields in transport measurements for metal thin films/ TbMn_6Sn_6 , and 2) an enhancement of conductance in point-contact conductance spectra for single TbMn_6Sn_6 . The authors point out that the emergent superconductivity could be unconventional; however, in my opinion, their experimental data are NOT enough and solid to support their idea. In this sense, the paper is NOT suitable for publication in Nature Communication. The issues have been listed as follows:

Overall Response:

We are grateful for Reviewer #3's valuable suggestions and comments to improve the manuscript. The concerns of the reviewer are mainly on three issues:

(A) It is suspected that the emergent superconductivity may result from the possible superconducting filamentary/impurities induced by thin film coating and mechanical force.

(B) Whether the properties of the TbMn_6Sn_6 are inherited in the degraded layer.

(C) 'the authors need to clarify the origin of the emergent superconductivity, which is acceptable in parallel for both systems: metal thin film/ TbMn_6Sn_6 and point-contact/ TbMn_6Sn_6 '.

Our overall responses are the following:

(A) To rule out the possibility that the superconducting signal results from superconducting filamentary/impurities induced by thin film coating and mechanical force, we conducted further TEM, EDS, and transport measurements. Near the surface of the as-grown TbMn_6Sn_6 sample and $\text{TbMn}_6\text{Sn}_6/\text{Au}$ heterostructure, similar tin-deficient layers with consistent element distribution are observed by EDS results, demonstrating no superconducting filamentary/impurities exist in the TbMn_6Sn_6 capped with the metal film. In addition, the possible influence of mechanical force is excluded by the emergent superconductivity in a $\text{TbMn}_6\text{Sn}_6/10$ nm Au heterostructure (s14), of which the TbMn_6Sn_6 sample surface is only ultrasonic cleaned by ethanol and acetone. Furthermore, the quasi-2D superconductivity of $\text{TbMn}_6\text{Sn}_6/\text{Au}$ heterostructure

is revealed by angle-dependent transport experiments, which demonstrate the emergent superconductivity originates from the 2D interface between the TbMn₆Sn₆ and deposited metal film, rather than superconducting filamentary/impurities.

(B) To determine the crystalline properties of the degraded layer, we carried out more detailed studies of the crystal structure of the degraded layer near the TbMn₆Sn₆/Au interface. Although the content of the Sn element is relatively low near the interface, the interplanar lattice spacing values in the degraded layer are still consistent with that of TbMn₆Sn₆ bulk crystal, suggesting that the degraded layer is mainly composed of polycrystalline TbMn₆Sn₆. Consequently, the properties of the TbMn₆Sn₆, such as the magnetization and strong spin-orbit coupling, are expected to be inherited in the degraded layer.

(C) The structural and elemental analyses at the interface indicates that the degraded TbMn₆Sn₆ layer may play an essential role in the emergence of superconductivity. To further understand the origin of degraded layers, we performed new TEM measurements on the as-grown TbMn₆Sn₆ samples. The TEM results show that the as-grown TbMn₆Sn₆ also has a degraded layer near the sample surface, indicating that the degraded layer already exists before the metal film deposition or metallic tip contact measurements. Therefore, when a metallic tip contacts the surface of TbMn₆Sn₆, the interface situation is similar to the TbMn₆Sn₆/metal film heterostructure, thus the emergent superconductivity can be observed in both systems.

Below we provide point-to-point responses to Reviewer #3's comments, which have been incorporated into our revised manuscript. We hope Reviewer #3 will find the reply and revision satisfactory.

Comment:

1) Since zero-resistance has not been observed in the transport measurement, one surely can suspect the formation of superconducting filamentary or impurities in TbMn₆Sn₆ (or in the degraded layer) by the thin film coating and point contact (i.e., due to mechanical force). Based on this suspicion, many of their interpretations are not compelling. For example, from the hysteretic magnetoresistance (MR) behavior, the authors suggest the superconductivity coupled with the magnetization of TbMn₆Sn₆, but it could be independent of each other. The magnetic layer is also still underneath.

Response: We thank Reviewer #3 for raising this comment. As pointed out in Fig. 4

and the text of our manuscript, the elemental mapping results near the surface of TbMn_6Sn_6 capped with Au film show a tin-deficient degraded layer, which excludes the existence of superconducting tin filamentary at the interface. Under the degraded layer, the TEM image represents a regular TbMn_6Sn_6 lattice structure that does not contain any superconducting filamentary too. To further confirm the existence of a tin-deficient degraded layer, we carried out TEM and EDS measurements on another $\text{TbMn}_6\text{Sn}_6/\text{Au}$ heterostructure (s17). As shown in Fig. R4, a similar tin-deficient degraded layer is observed between the regular TbMn_6Sn_6 lattice structure and capping metal film, suggesting that no extra superconducting filamentary is induced in the $\text{TbMn}_6\text{Sn}_6/\text{Au}$ heterostructure during the heterostructure fabrication.

Fig. R4 | Structural image and elemental mappings of $\text{TbMn}_6\text{Sn}_6/\text{Au}$ heterostructure (s17). **a-b** The cross-sectional HAADF STEM images of the TbMn_6Sn_6 single crystal and the $\text{TbMn}_6\text{Sn}_6/\text{Au}$ heterostructure. The heterostructure is fabricated by depositing 10 nm Au film on the (001) surface of the TbMn_6Sn_6 single crystal. **c-f** The EDS mapping of Au, Tb, Mn, and Sn elements in the HAADF STEM image **b**. The interface between the TbMn_6Sn_6 and the Au film is marked by a green dashed line. The degraded TbMn_6Sn_6 layer is marked by a yellow dashed line box.

Further TEM and EDS measurements demonstrate that the tin-deficient degraded layer is naturally formed in the as-grown non-superconducting TbMn_6Sn_6 sample rather than due to the heterostructure fabrication process. Fig. R2 shows a typical high-angle annular dark-field STEM (HAADF STEM) image and the elemental mapping results

of an as-grown TbMn_6Sn_6 sample, which is only ultrasonic cleaned by ethanol and acetone. Similar to the $\text{TbMn}_6\text{Sn}_6/\text{Au}$ heterostructures, a tin-deficient degraded layer exists near the sample surface.

Fig. R2 | Structural image and elemental mappings of the as-grown TbMn_6Sn_6 single crystal. **a** A cross-sectional HAADF STEM image of the TbMn_6Sn_6 single crystal surface. **b-d** The EDS mappings of Tb, Mn, and Sn elements distribution of the HAADF STEM image **a**. The interface between the TbMn_6Sn_6 and the shielding layer, a carbon protecting layer with the thickness of more than 100 nm, is marked by a green dashed line. The degraded TbMn_6Sn_6 layer is marked by a yellow dashed line box.

The tin-deficient property is also confirmed by EDS measurements equipped in the scanning electron microscope, which show the chemical composition ratio of the as-grown TbMn_6Sn_6 sample and $\text{TbMn}_6\text{Sn}_6/\text{Au}$ heterostructure near the surface. More than twenty regions at different locations on each sample surface are measured, and all the measured areas show a substantially lower concentration of Sn element (Sn: Tb \sim 4.66:1

and typical results can be found in Fig. R5) than the nominal composition of single-crystalline TbMn_6Sn_6 (Sn: Tb \sim 6:1), confirming no superconducting tin filamentary near the sample surface.

Fig. R5 | Energy-dispersive X-ray spectroscopy (equipped in the scanning electron microscope) results of TbMn_6Sn_6 as-grown sample (a) and TbMn_6Sn_6 capped 10 nm Au on the (001) surface (b).

After determining the existence of a tin-deficient degraded layer in both the as-grown TbMn_6Sn_6 sample and $\text{TbMn}_6\text{Sn}_6/\text{Au}$ heterostructure, we carried out transport measurements on an as-grown TbMn_6Sn_6 sample (s14) with and without capping the Au film. The surface of s14 is only ultrasonic cleaned by ethanol and acetone to avoid the possible influence of mechanical force. As shown in Fig. R6, no superconducting transition is detected in the as-grown TbMn_6Sn_6 sample (s14) without capping Au film, indicating that there is no superconducting filamentary or impurity in the as-grown TbMn_6Sn_6 sample with the degraded layer; superconductivity is emergent when a 10 nm Au film is capped on the same TbMn_6Sn_6 sample (s14). Therefore, the emergent superconductivity is expected to originate from the interaction between the TbMn_6Sn_6 surface and the deposited metal film.

Fig. R6 | The normalized resistance vs temperature curves of TbMn₆Sn₆ s14 (a) and TbMn₆Sn₆ s14 capped 10 nm Au on the (001) surface (b).

More importantly, motivated by the reviewer’s comment, we have carried out transport experiments to reveal the anisotropy of the emergent superconductivity at the TbMn₆Sn₆/metal heterostructure. As shown in Fig. R7, a cusp-like peak is clearly observed in the angular dependence of the critical magnetic field $B_c(\theta)$ for the TbMn₆Sn₆/Au heterostructure, here θ is the angle between the magnetic field and c -axis of TbMn₆Sn₆ sample. The peak feature can be well fitted by the 2D Tinkham model and deviates from the 3D anisotropic mass model, demonstrating a quasi-2D superconductivity at the TbMn₆Sn₆/Au heterostructure. Moreover, the temperature-dependence of the critical field presented in Fig. R7c and Fig. R7d show the $(T_c - T)^{1/2}$ -dependence of $B_{c//}(T)$ (magnetic field applied perpendicular to c -axis) and T -linear dependence of $B_{c\perp}(T)$ (magnetic field applied along c -axis) near T_c , respectively, which are also consistent with the phenomenological 2D Ginzburg-Landau (G-L) model, further supporting the quasi-2D superconductivity and excluding the possible influence of the “superconducting filamentary” or “impurities”.

Fig. R7 | Quasi-2D superconductivity in the TbMn₆Sn₆/Au heterostructure. **a** Angular dependent magnetoresistance at 2 K. Here θ is the angle between the magnetic field and c -axis of TbMn₆Sn₆ sample. **b** Angular dependence of the critical magnetic field $B_c(\theta)$. Inset shows the zoom-in of the $B_c(\theta)$ around 90°. The blue curve is the fitting with 2D Tinkham model $(B_c(\theta)\sin(\theta)/B_{c//})^2 + |B_c(\theta)\cos(\theta)/B_{c\perp}| = 1$ and the red curve is the fitting with the 3D anisotropic mass model $B_c(\theta) = B_{c//}/(\sin^2(\theta) + \gamma^2\cos^2(\theta))^{1/2}$ with $\gamma = B_{c//}/B_{c\perp}$. **c** The temperature dependence of the critical magnetic field $B_{c//}(T)$ under the magnetic field applied perpendicular to the c -axis near T_c . The red curve is the 2D G-L fitting. The inset shows the temperature dependence of normalized resistance under different magnetic fields applied perpendicular to the c -axis. **d** The temperature dependence of the critical magnetic field $B_{c\perp}(T)$ under the magnetic field applied along the c -axis near T_c . The red curve is the 2D G-L fitting. The inset shows the temperature dependence of normalized resistance under different magnetic fields applied along the c -axis. B_c in **b-d** is defined as the magnetic field corresponding to 95% normal resistance.

As for the superconductivity and ferromagnetism coupling, we would like to point out that apart from the magnetoresistance hysteresis in the superconducting state, the superconducting signal characterized by the resistance drop becomes more notable after the magnetization treatment on the TbMn₆Sn₆/Au heterostructure (Fig. R8). The enhanced superconductivity by magnetization indicates that magnetism is favorable for the formation of superconductivity and thus the superconductivity is expected to be coupled to magnetism.

Fig. R8 | The normalized resistance vs temperature curves of TbMn₆Sn₆ (s10) capped 10 nm Au on the (001) surface after magnetization and demagnetization treatment.

Comment:

2) The authors suggest the degraded layer is superconducting, but there is no reason to have inherent magnetization and strong SOC from TbMn₆Sn₆ because it is no longer to be TbMn₆Sn₆. They need to secure specific information for the degraded layer, like STM or ARPES measurement, to suggest unconventional superconductivity. Isn't it possible to remove the metal thin film atop and perform other measurements?

Response: We thank Reviewer #3 for this helpful suggestion. Inspired by Reviewer #3's suggestion, we have conducted a more in-depth study of the degraded layer in the TbMn₆Sn₆/Au heterostructure. We further analyzed the lattice structures in the degraded layer of the TbMn₆Sn₆/Au heterostructure. In the degraded layers of two typical TbMn₆Sn₆/Au heterostructures s8 (shown in Fig. 4 in the main text) and s17, we can identify some regions showing lattice structures that are consistent with the crystal structure of TbMn₆Sn₆. Fig. R1 displays three high-angle annular dark-field STEM (HAADF STEM) images of the degraded layers in s8 (Fig. R1a) and s17 (R1b-c). These images present regions showing crystal structures in the degraded layer, and the interplanar lattice spacing values in these regions are approximately 4.5 Å. Using the PDF Card (NO. 01-072-3490) as a reference, this value corresponds to the space distance of (002) TbMn₆Sn₆ along the [001] crystal orientation (4.502 Å). These findings suggest that the degraded layer mainly consists of the polycrystalline TbMn₆Sn₆. Hence, the degraded layer inherits the magnetization and strong SOC from TbMn₆Sn₆, and the interface superconductivity could be unconventional.

In addition, as mentioned above, the interface superconductivity is sensitive to the

magnetic properties of the TbMn_6Sn_6 sample. As shown in Fig. R8, the superconducting signal characterized by the resistance drop becomes more notable after the magnetization treatment on the $\text{TbMn}_6\text{Sn}_6/\text{Au}$ heterostructure. This also indicates that ferromagnetism positively promotes the generation of superconductivity, and further supports the coupling of ferromagnetism and superconductivity.

Fig. R1 | Further analyses of the microstructure of the degraded TbMn_6Sn_6 layer near the interface of two typical $\text{TbMn}_6\text{Sn}_6/\text{Au}$ heterostructures s8 (a) and s17 (b-c). The $\text{TbMn}_6\text{Sn}_6/\text{Au}$ heterostructures are fabricated by depositing 10 nm Au film on the (001) surface of the TbMn_6Sn_6 single crystal. The selected area is marked by white squared frames. **a-c** Cross-sectional HAADF STEM images of the crystal lattice in the degraded TbMn_6Sn_6 layer near the interface of $\text{TbMn}_6\text{Sn}_6/\text{Au}$ heterostructure s8 (a) and s17 (b-c). The interplanar spacing values of the lattices are marked by two red lines. The white arrows indicate the [001] orientation of the bulk TbMn_6Sn_6 single crystal.

In our revised manuscript, we have added the related discussion and analyses in the main text and Supplementary Information.

Comment:

3) More systematic PCS measurements need to be carried out, and more detailed information on their measurements should be provided. According to their interpretation, the mechanical force to make a point contact is expected to lead to a local degradation and superconductivity on the surface of TbMn_6Sn_6 , and the degradation would be worse with increasing the applied force, which could be estimated from contact resistance. The contact resistance is not directly indicative of the diameter of a point contact — actual contact can consist of many parallel contacts due to the microstructures of a tip and sample.

4) Along with comment 3), the authors need to clarify the origin of the emergent superconductivity, which is acceptable in parallel for both systems: metal thin film/ TbMn_6Sn_6 and point-contact/ TbMn_6Sn_6 . I don't think the point contact does make such a huge degradation (e.g., stoichiometry change) as observed in the TEM and EDX measurements on the $\text{Au}/\text{TbMn}_6\text{Sn}_6$.

Response: Let us address Comments 3) and 4) together, as they both relate to the same

issue. As mentioned above, in response to Reviewer #3's comments, we conduct TEM (Transmission electron microscopy) and EDS (Energy-dispersive X-ray spectroscopy) studies on the as-grown TbMn₆Sn₆ single crystal (Fig. R2). As shown in Fig. R2, similar to the TEM data of Au/TbMn₆Sn₆ heterostructure shown in Fig. 4 in the manuscript, the TEM image of the as-grown TbMn₆Sn₆ also shows the degraded layer at the sample surface. Therefore, the degraded layer already exists before the metal film (Au, Ag, and Ni) deposition or point contact measurements, which explains why both metal thin film/TbMn₆Sn₆ and point-contact/TbMn₆Sn₆ show similar interface superconductivity.

We agree with the comments ‘The contact resistance is not directly indicative of the diameter of a point contact — actual contact can consist of many parallel contacts due to the microstructures of a tip and sample’. The related statements have been removed in the revised manuscript.

In our revised manuscript, we have added the related discussion in the main text and Supplementary Information.

Comment:

5) In the conductance spectra, significant dips are consistently observed, which may narrate the thermal regime or so-called Maxwell component of the point contact resistance. In such a regime, the conductance spectra may reflect temperature dependence resistance, not only spectroscopic information. The observation of the zero-bias conductance peak-like feature (in SI) could also be due to the same reason. They need to revisit the contact resistances and provide information on those.

Response: We thank Reviewer #3 for this reminder. As Reviewer #3 points out, ‘the contact resistance is not directly indicative of the diameter of a point contact — actual contact can consist of many parallel contacts due to the microstructures of a tip and sample.’ We cannot determine which regime the point contact belongs to based on the contact resistance alone in the superconducting point contact measurements. Normally, in the thermal regime, the conductance spectra may reflect temperature dependence resistance and show the zero-bias conductance peak (ZBCP) in the point contact spectra (PCS). However, in such a regime, the full width at half maximum (FWHM) of the ZBCP usually increases as the temperature decreases, which is opposite to our observations shown in Supplementary Fig. 17b-d (the observed ZBCP becomes sharper and higher at lower temperatures). Related discussions have been added in the revised version of Supplementary Information.

Comment:

6) The authors need to be more careful in preparing the figures in the manuscript and SI. For example:

The tick direction is not consistent (Figure 2 and Figure 3).

The scale bars are too small and not visible (Figure 4)

No x-axis (top and bottom) (Supplementary Figure 2, Figure 8, Figure 12 (a) and (b), probably crop error)

No label for supplementary Fig. 12 (a, b, d, e).

Response: We thank Reviewer #3 for the reminder. The tick directions and scale bars have been revised as suggested by Reviewer #3. Furthermore, we have carefully checked the manuscript we submitted and found that the missing information mentioned by the reviewer was not missing. Therefore, the missing information in the reviewer's version may be a software problem or caused by the online reviewer system for this manuscript.

Comment:

7) The authors need to add more information on the fabrication of the metal thin films, like a method, deposition temperature, Etc. They may find a deposition condition that would not lead to surface degradation.

Response: We thank the reviewer for this suggestion. As we have discussed above, the surface degradation is not induced by the deposition condition. Even though, we have added a more detailed fabrication process of the metal film/ TbMn₆Sn₆ surfaces into the Method section:

Before preparing the TbMn₆Sn₆/metallic film heterostructures, pre-treating processes were applied to the TbMn₆Sn₆ samples, including mechanically cleaving (Fig. 2a-b), surface polishing (Fig. 2c-d, Supplementary Fig. 2 and Fig. 5), or only ultrasonic cleaning by ethanol and acetone (Fig. 2e-f, Supplementary Fig. 4 and 14). Subsequently, the samples were transferred to the e-beam evaporation chamber of an LJUHV E-400L E-Beam Evaporator. The 10 nm-thick Au (or 8 nm-thick Ni) films were deposited onto the sample surface at room temperature by the standard e-beam evaporation method with the electron energy of 8 keV. The chamber pressure is lower than 2×10^{-6} Torr before deposition and was maintained below 4×10^{-6} Torr during the evaporation

process. The purities of the commercial Au and Ni targets are 99.999%, and a deposition rate of 0.5 Å/s (0.3 Å/s) was achieved by focusing the electron beam on the Au (Ni) target. The 10-nm thick Ag film in Supplementary Fig. 6 was grown at room temperature by using DC reactive magnetron sputtering method with the MSP-3200 Magnetron Sputtering System. In the sputtering chamber, the Ag film (target impurity, 99.99%) was deposited onto the TbMn₆Sn₆ surface at a rate of 2.7 Å/s with an Ar gas pressure of 0.9 Pa.

Reviewers' Comments:

Reviewer #1:

Remarks to the Author:

The authors have revised their manuscript with additional data (TEM analyses), which strongly supports the authors conclusion. I am satisfied with their revision.

Reviewer #3:

Remarks to the Author:

The authors have dedicated significant effort to address the comments and questions raised by the reviewers. However, I have noticed that many questions still need to be clearly addressed regarding the origin of superconductivity and its topological nontriviality. I agree that their transport and PCS data suggest the formation of the superconducting phase. However, the significance and novelty of their work need to be suggested by the emergence of possible topological nontrivial superconductivity. This manuscript may not yet meet the necessary criteria for publication in Nature Communications. I request that the authors carefully consider the detailed and specific questions listed below.

1) The author tried to rule out the formation of superconducting filamentary or impurity by comparative analysis of the samples with (Fig. 4 and Fig. R4) and without (Fig. R2 and Supp. Fig. 13) Au layer on the top. In Fig. R4 of their response, however, the Sn deficiency layer exists deep underneath the Au/TbMn₆Sn₆ interface, which differs from the EDX map of the single crystal. I have also noticed that the samples used for the transport measurements (s7, s15, and s10) and structural measurements (s8 and s17) are different. As sample preparation issues are raised in the other materials, surface polishing carried out in the study needs to be considered carefully – the surface degradation might be attributed to the surface polishing.

2) The formation of the degraded layer seemed to be attributed to the emerged superconductivity in the previous manuscript, but now the metallic layer or contact seems to be emphasized according to the response. What is the role of the metal layer or contact? The first paragraph of the discussion part (page 9, line 298) reads, "A plausible mechanism is that the carrier doping may take place near the interface and enable an emergent quasi-2D superconductor" The authors need to provide experimental evidence.

3) (Extended from the previous comment 2) The "partial" presence of polycrystalline TbMn₆Sn₆ cannot be the evidence that the emergent interface superconductivity is expected to inherit the magnetization and strong SOC from the TbMn₆Sn₆. The local TEM measurement, including HAADF-STEM, does not guarantee that the degraded layer is mostly polycrystalline TbMn₆Sn₆ despite such a huge Sn deficiency as observed in the EDX mapping. Furthermore, the inter-planar distance measured by HAADF-STEM is insufficient to support that, especially for such 2D materials. Their concern seems only focused on the possible formation of superconducting Sn phases. Any superconducting phase other than Sn can be formed somehow.

4) If an interfacial superconducting phase emerges somehow without the strong evidence of the topologically nontrivial superconductivity, what is the difference with just conventional superconductor (Nb, Al, or Sn)/TbMn₆Sn₆ structures? What is the uniqueness and advantage of their system? The data provided to support the coupling of FM and superconductivity is only the supplementary Fig. 3.

5) In the previous comment 3, I asked them to provide more systematic PCS measurements and contact resistance information, but they did not provide such information and/or additional data. How could they observe the ZBCP feature in PCS measurement with only the Au tip (Supplementary Figure 17)? – what is the difference with the other measurements? They need to provide contact resistance for all the PCS measurements. And also, the author said, "this behavior cannot be explained by the mechanism of point contact in thermal regime ..." (page 7, line 161 in SI), but no clear explanation or reference has not been provided. In addition, I was not able to find

PCS results with soft contacts.

6) To validate their PCS results, they may try BTK fits to their dI/dV spectra. Is a superconducting gap estimated from the dI/dV spectra consistent with the other data?

7) (Minor) In the previous comment 1, I pointed out that the zero-resistance is not observed. A simple explanation could be the inhomogeneity of the superconducting phase, but they need to discuss that in the manuscript. (Related to the response to Reviewer #2's comment ii) Is it possible to be related to the "partial" (non-superconducting) polycrystalline $TbMn_6Sn_6$ and the rest (superconducting but not-identified phase)?

8) (Minor) In the abstract, "two-dimensional semiconductors proximity-coupled to s-wave superconductors and insulating ferromagnets" – this seems not to reflect their work.

9) (Minor) double conductance peaks combined with double conductance dips in the PCS for the case in the intermediate regime – this is the wrong expression. And the double conductance enhancement can be observed when $Z=0$ (no barrier)

10) (Minor) In the response letter, I suggest they provide revised content to each comment if they have revised the manuscript according to the reviewers' comments.

The reviewers' original comments are shown in blue characters. Our responses are shown in black characters, and the revised contents made to the manuscript are shown in purple characters.

REVIEWER COMMENTS

Reviewer #1 (Remarks to the Author):

The authors have revised their manuscript with additional data (TEM analyses), which strongly supports the authors conclusion. I am satisfied with their revision.

Response

We are sincerely grateful to Reviewer #1 for the insightful comments in the previous round of the review and the recommendation to publish our manuscript in *Nature Communications*.

Reviewer #3 (Remarks to the Author):

Comment:

The authors have dedicated significant effort to address the comments and questions raised by the reviewers. However, I have noticed that many questions still need to be clearly addressed regarding the origin of superconductivity and its topological nontriviality. I agree that their transport and PCS data suggest the formation of the superconducting phase. However, the significance and novelty of their work need to be suggested by the emergence of possible topological nontrivial superconductivity. This manuscript may not yet meet the necessary criteria for publication in *Nature Communications*. I request that the authors carefully consider the detailed and specific questions listed below.

Response

We are grateful for Reviewer #3's recognition of our efforts, which are also appreciated by Reviewer #1 by saying that "*The authors have revised their manuscript with additional data (TEM analyses), which strongly supports the authors conclusion*". We also sincerely thank Reviewer #3 for supporting our observation of the superconducting phase in the TbMn₆Sn₆/metal heterostructure.

In this round of review, Reviewer #3 raised some comments on the significance of our work. Actually, the observation of emergent superconductivity in the TbMn₆Sn₆/metal heterostructure, which is supported by Reviewer #3, is a novel finding since the observed superconductivity suggests that the topological magnet and metal heterostructure could form a new class of interface superconductors. As highly evaluated by Reviewer #2, the emergent superconductivity in TbMn₆Sn₆ "*is an impressive discovery*". Furthermore, the emergent interface superconductivity in the TbMn₆Sn₆/metal heterostructure provides a promising platform for exploring

topological superconductivity. As pointed out by Reviewer #1, the emergent superconductivity in the TbMn₆Sn₆/metal heterostructure is “*a very interesting result because it may be the long-term-awaited time-reversal symmetry-breaking spin-triplet superconductivity*” and Reviewer #2 also give a high evaluation by saying “*this is a high-quality research paper for exploring new superconductors and chiral topological superconductivity via the interface engineering of the heterostructure composed of magnetic topological materials and non-superconducting metals*”. Based on our experimental findings, Reviewer #1 and Reviewer #2 concluded that our work will be “*highly appreciated by the condensed matter community*” and “*A broad audience could be inspired by this work*”, respectively. Therefore, both of them recommend the publication of our paper in *Nature Communications*.

As for Reviewer #3’s concerns about the topological nontriviality of the emergent superconductivity in TbMn₆Sn₆/metal heterostructure, we would like to point out that compelling evidence of topological superconductivity has not been obtained in any systems yet. Although we also cannot fully confirm the topological superconductivity based on our current experimental results, the significance and novelty of our work can be demonstrated from two aspects (briefly discussed in the previous paragraph).

1. As mentioned in our manuscript, TbMn₆Sn₆ is a kagome magnet with topological non-trivial electronic bands and a Curie temperature (T_{curie}) of 423 K. Unexpectedly, the quasi-two-dimensional (2D) superconductivity is induced by depositing metallic thin films on the TbMn₆Sn₆ surface, which is the main observation in our work. The interface superconductivity generated between the topological magnetic material and non-superconducting metal reveals a novel class of interface superconductivity systems. As highly evaluated by Reviewer #2 by saying “*It is an impressive discovery that superconductivity can emerge in TbMn₆Sn₆ single crystals*”.

2. Theoretically, the topological superconductivity can be induced by stacking an *s*-wave superconductor, a 2D semiconductor with Rashba spin-orbit coupling (SOC), and a ferromagnetic insulator to form a heterostructure. However, this proposal has not been experimentally realized yet. In our work, the quasi-2D superconductivity is induced in the TbMn₆Sn₆/metal heterostructure. The strong SOC can be offered by both the TbMn₆Sn₆ part (e.g. heavy Tb atoms) and the metallic film part near the interface. Our experimental results indicate the observed interface superconductivity is coupled to the ferromagnetism (detailed discussions are shown in the point-to-point response). Therefore, the TbMn₆Sn₆/metal heterostructure is an experimental realization in excellent agreement with the theoretical strategy for topological superconductivity and provides a new promising platform for exploring topological superconductivity. As Reviewer #1 said ‘*it may be the long-term-awaited time-reversal symmetry-breaking spin-triplet superconductivity.*’ Furthermore, Reviewer #2 pointed out that our work “*is a high-quality research paper for exploring new superconductors and chiral topological superconductivity*”.

Below we provide point-to-point responses to Reviewer #3's comments, which have been incorporated into our revised manuscript. We hope Reviewer #3 will find the reply and revision satisfactory.

Comment:

1) The author tried to rule out the formation of superconducting filamentary or impurity by comparative analysis of the samples with (Fig. 4 and Fig. R4) and without (Fig. R2 and Supp. Fig. 13) Au layer on the top. In Fig. R4 of their response, however, the Sn deficiency layer exists deep underneath the Au/TbMn₆Sn₆ interface, which differs from the EDX map of the single crystal. I have also noticed that the samples used for the transport measurements (s7, s15, and s10) and structural measurements (s8 and s17) are different. As sample preparation issues are raised in the other materials, surface polishing carried out in the study needs to be considered carefully – the surface degradation might be attributed to the surface polishing.

Response

We thank the reviewer for these comments. Although the elemental mappings of the TbMn₆Sn₆/Au interface (Fig. 4, Fig. R4 in the last round of review) and the single crystal (Supplementary Fig. 14 in the revised version) are slightly different in some details, all of them show Sn-deficient layer near the surface of the TbMn₆Sn₆, which exclude the possibility of superconducting Sn filamentary.

The reviewer mentioned that the samples used for the transport measurements (s7, s15, and s10) and structural measurements (s8 and s17) are different. Actually, the transport results of s17 have been shown in the last round of response letters as Fig. R3 (also shown in Fig. R1 in this round of review). As shown in Fig. R1, the temperature-dependent resistance curves of s17 show notable superconducting drops and the superconducting transition temperature is about 3.8 K, consistent with the results of other heterostructures shown in our manuscript. After observation of the emergent superconductivity, the TEM measurements were performed on s17 to obtain the structural and elemental mapping information of the same superconducting structure (Fig. R4 in the last round of review). Furthermore, the TbMn₆Sn₆ samples used in TbMn₆Sn₆/Au heterostructure s7 (for transport) and s8 (for TEM) are obtained by mechanically cleaving one TbMn₆Sn₆ crystal into two parts. The fabrication processes of heterostructures s7 and s8 are identical. Therefore, the samples s7 and s8 can be taken as the same sample. The transport and TEM measurements on the same sample (s7, s8, and s17) make our observation of the emergent superconductivity in TbMn₆Sn₆/Au heterostructures and the related analyses more reliable. We have further emphasized the information in the revised manuscript.

In the revised manuscript, we added some contents as below:

In the ‘Sample treatment and film deposition’ of the ‘Methods’ section: The TbMn₆Sn₆ samples used in TbMn₆Sn₆/Au heterostructure s7 and s8 are obtained by mechanically cleaving one TbMn₆Sn₆ crystal into two parts. The fabrication processes of heterostructures s7 and s8 are identical.

Fig. R1 | Temperature dependence of the normalized longitudinal resistance of TbMn₆Sn₆/Au heterostructure s17.

As for the reviewer's concerns about the influence of surface polishing on the formation of the degraded layer, we have noted that TbMn₆Sn₆ samples used in s7 and s8 are obtained by mechanically cleaving one TbMn₆Sn₆ crystal into two parts without a polishing process. Furthermore, the TbMn₆Sn₆ samples used in heterostructures shown in the last round of review (s10, s14, s17) are only ultrasonic cleaned by ethanol and acetone to avoid the possible influence of surface polishing. The main observations in our work, including the emergent superconductivity and tin-deficient degraded layer, are shown in these no-surface-polishing samples. We have further clarified the information in the revised manuscript.

In the revised manuscript, we added some contents as below:

1. Captions of Supplementary Fig. 13: The R - T curves of s17 can be referred to Supplementary Fig. 19. The TbMn₆Sn₆ sample used in s8 is obtained by mechanically cleaving without a polishing process. The TbMn₆Sn₆ sample used in s17 is only ultrasonic cleaned by ethanol and acetone without surface polishing.
2. Supplementary Fig. 19:

Supplementary Fig. 19 | Temperature dependence of the normalized longitudinal resistance of the TbMn₆Sn₆/Au heterostructure s17.

Comment:

2) The formation of the degraded layer seemed to be attributed to the emerged superconductivity in the previous manuscript, but now the metallic layer or contact seems to be emphasized according to the response. What is the role of the metal layer or contact? The first paragraph of the discussion part (page 9, line 298) reads, “A plausible mechanism is that the carrier doping may take place near the interface and enable an emergent quasi-2D superconductor” The authors need to provide experimental evidence.

Response

The degraded layer, which is naturally formed near the surface of as-grown TbMn_6Sn_6 , is non-superconducting. Thus, the emerged superconductivity in the interface between TbMn_6Sn_6 and non-superconducting metal cannot be simply attributed to the formation of the degraded layer. The metallic layer may play an important role in the emergence of topological nontrivial 2D superconductivity. We elucidate this from the following two aspects.

Firstly, our experimental results demonstrate that superconductivity emerges when the surface of TbMn_6Sn_6 single crystals is coated with a metallic thin film. To explain the emergent superconductivity, we proposed a possible scenario that the carrier transfer between the metallic film or contact and the TbMn_6Sn_6 near the interface may occur and lead to superconductivity. Besides, in this scenario, the metallic layer with SOC may provide the band with a nonzero Berry phase, and when it is proximitized to the s -wave superconductor, the topological superconductivity can emerge.

Secondly, in addition to the case mentioned above, we emphasized that the Rashba-type SOC induced by the metallic film depositing on the TbMn_6Sn_6 surface might also play a role in the formation of superconductivity. As discussed in Supplementary Text III and IV, the large Rashba-type SOC can mix the spin-up and spin-down components of the electrons in TbMn_6Sn_6 . Such mixing of spin components induced by the metallic layer overcomes the magnetization from the TbMn_6Sn_6 that may suppress the superconducting pairing effect. Consequently, the formation of superconductivity becomes possible.

Both scenarios can lead to interface superconductivity with nontrivial topology, and in each scenario the metallic layer is necessary. Since the main focus of our work is the interface superconductivity generated between the topological magnetic material and non-superconducting metal, further determining the mechanism of emergent superconductivity from a more fundamental level is beyond the scope of our work. Nevertheless, that does not affect the importance of our experimental discovery which would definitely stimulate more experimental and theoretical investigations to understand such interesting interface emergent superconductivity.

Comment:

3) (Extended from the previous comment 2) The “partial” presence of polycrystalline TbMn_6Sn_6 cannot be the evidence that the emergent interface superconductivity is expected to inherit the magnetization and strong SOC from the TbMn_6Sn_6 . The local TEM measurement, including HAADF-STEM, does not guarantee that the degraded layer is mostly polycrystalline TbMn_6Sn_6 despite such a huge Sn deficiency as observed in the EDX mapping. Furthermore, the inter-planar distance measured by HAADF-STEM is insufficient to support that, especially for such 2D materials. Their concern seems only focused on the possible formation of superconducting Sn phases. Any superconducting phase other than Sn can be formed somehow.

Response

We thank the reviewer for the comments. Actually, in the last round of review, Reviewer #1 was also concerned that the degraded layer ‘*does not have the crystal structure of TbMn_6Sn_6 , then all discussions of superconductivity at the Au/ TbMn_6Sn_6 interface are meaningless*’. By performing further HAADF-STEM measurements, we determined the presence of polycrystalline TbMn_6Sn_6 in the degraded layer by the inter-planar distance, a common method to obtain the crystallinity based on the current TEM technique [*Adv. Mater.* **15**, 1022-1025 (2003); *Nano. Lett.* **20**, 3611-3619 (2020)]. These results removed Reviewer #1’s concern. As Reviewer #1 pointed out, our data ‘*strongly supports the authors’ conclusion*’.

As Reviewer #3 recognized, the polycrystalline TbMn_6Sn_6 exists in the degraded layer near the interface. Therefore, it is natural to expect the SOC and magnetization of polycrystalline TbMn_6Sn_6 can be inherited by the interface superconductivity. Furthermore, the heavy Tb atoms near the interface, as shown in the elemental mappings, also provide a strong SOC that can be inherited by the superconductivity. As for the coupling of superconductivity and ferromagnetism, we would like to point out that TbMn_6Sn_6 has a ferrimagnetic ground state with T_{curie} of 423 K. Consequently, the emergent superconductivity in TbMn_6Sn_6 /metal heterostructure is naturally compatible with ferromagnetism. Furthermore, our experimental results support the coupling of superconductivity and ferromagnetism from the following aspects: i) The magnetoresistance shows a hysteresis in the superconducting state, which suggests the emergent superconductivity coexists with ferromagnetism, as shown in Fig. 2b and Fig. 3d. ii) the statistics of T_c versus the tip materials (see Supplementary Fig. 11b) show no notable variations in the maximum T_c values by paramagnetic (Au, PtIr) or ferromagnetic (Ni) tips. This implies that the emergent superconductivity can tolerate ferromagnetism. iii) The resistance drop becomes more noticeable after the magnetization treatment on the TbMn_6Sn_6 /Au heterostructure (Supplementary Fig. 3), indicating that magnetism is favorable for the formation of superconductivity.

To the best of our knowledge, in our samples, other elements or compounds besides Sn cannot exhibit superconductivity. Thus, we made an effort to exclude the existence of superconducting Sn filaments near the interface. The observation of quasi-2D

superconductivity also excludes other possibilities besides the emergent interface superconductivity.

Comment:

4) If an interfacial superconducting phase emerges somehow without the strong evidence of the topologically nontrivial superconductivity, what is the difference with just conventional superconductor (Nb, Al, or Sn)/TbMn₆Sn₆ structures? What is the uniqueness and advantage of their system? The data provided to support the coupling of FM and superconductivity is only the supplementary Fig. 3.

Response

As mentioned in our manuscript, TbMn₆Sn₆ is a kagome magnet with topological non-trivial electronic bands and a Curie temperature of 423 K. Unexpectedly, the quasi-2D superconductivity is induced by depositing metallic thin films on the TbMn₆Sn₆ surface, which is the main observation in our work. The interface superconductivity generated between the topological magnetic material and non-superconducting metal reveals a novel class of interface superconductivity systems, which demonstrates the uniqueness of the TbMn₆Sn₆/metal heterostructure and is highly evaluated by Reviewer #2 by saying “*It is an impressive discovery that superconductivity can emerge in TbMn₆Sn₆ single crystals*”.

Usually, the proximity effect is difficult to exist in materials with strong ferromagnetism. Hence, we are unsure whether the superconducting proximity effect can be realized in the conventional superconductor (Nb/Al/Sn)/TbMn₆Sn₆ heterostructure. Investigations on the proximity effect in TbMn₆Sn₆/conventional superconductor heterostructure might be interesting, but beyond the scope of our work. Moreover, until now, achieving topological superconductivity through the proximity effect remains a significant experimental challenge.

Compelling evidence of topological superconductivity has not been obtained in any system yet. Based on our current experimental results, we admit that it is hard to fully confirm the topological superconductivity. Theoretically, the topological superconductivity can be induced by stacking an *s*-wave superconductor, a 2D semiconductor with Rashba SOC, and a ferromagnetic insulator to form a heterostructure (Fig. 5a). However, this proposal has not been experimentally realized yet. In our work, the quasi-2D superconductivity is induced in the TbMn₆Sn₆/metal heterostructure. More importantly, strong SOC can be contributed by both TbMn₆Sn₆ containing heavy Tb atoms and metallic film near the interface. Our experimental results indicate the interface superconductivity is coupled to the ferromagnetism and thus likely to break time-reversal symmetry. Therefore, the TbMn₆Sn₆/metal heterostructure is similar to the theoretical proposal shown in Fig. 5a, which provides a new promising platform for exploring topological superconductivity. As Reviewer #1 said ‘*This is a very interesting result because it may be the long-term-awaited time-reversal symmetry-breaking spin-triplet superconductivity. The paper is highly appreciated by the condensed matter community.*’ Furthermore, Reviewer #2 pointed out that our work “*is a high-quality research paper for exploring new superconductors and chiral topological superconductivity via the interface engineering of the*

heterostructure composed of magnetic topological materials and non-superconducting metals”.

As for the experimental evidence supporting the coupling of ferromagnetism and superconductivity, we have provided detailed discussions in the response to Comment 3. As shown in our experimental results, the magnetoresistance shows a hysteresis in the superconducting state; there are no notable variations in the maximum T_c values by paramagnetic (Au, PtIr) or ferromagnetic (Ni) tips; and the superconductivity becomes more noticeable after the magnetization of the TbMn₆Sn₆/Au heterostructure (Supplementary Fig. 3). All these experimental results indicate that the emergent superconductivity is compatible with the ferromagnetism, and the scenario of topological nontrivial superconductivity is preferred in the TbMn₆Sn₆/metal heterostructure.

Comment:

5) In the previous comment 3, I asked them to provide more systematic PCS measurements and contact resistance information, but they did not provide such information and/or additional data. How could they observe the ZBCP feature in PCS measurement with only the Au tip (Supplementary Figure 17)? – what is the difference with the other measurements? They need to provide contact resistance for all the PCS measurements. And also, the author said, “this behavior cannot be explained by the mechanism of point contact in thermal regime ...” (page 7, line 161 in SI), but no clear explanation or reference has not been provided. In addition, I was not able to find PCS results with soft contacts.

Response

As summarized in the manuscript, we have done systematic point contact spectra (PCS) measurements: “we have probed 92 PC positions and 170 PC states on the different surfaces of two TbMn₆Sn₆ samples from the same batch using various metallic tips, including PtIr, Au, Ni, and Ag (see Supplementary Fig. 11a). All these non-superconducting tips are found to induce superconductivity on either the top (001) or side surfaces of TbMn₆Sn₆. The statistics of T_c versus the tip materials (see Supplementary Fig. 11b) show no significant variations in the maximum T_c values, whether the tip materials are relatively hard or soft, paramagnetic or ferromagnetic. These striking observations point to a universal and highly robust interface superconducting state in TbMn₆Sn₆/metal heterostructures.” As the reviewer mentioned in the last round of review, “*The contact resistance is not directly indicative of the diameter of a point contact — actual contact can consist of many parallel contacts due to the microstructures of a tip and sample*”. Since the point contact conduction regime (thermal, intermediate, or ballistic regime) is largely dependent on the contact size, the contact resistance cannot directly reflect the conduction regime. Similarly, the contact barrier cannot be determined by the contact resistance too. Therefore, we did not add the point contact resistance in our previous response. In our current revised manuscript, we have added the contact resistance information as suggested by the reviewer.

According to the reviewer’s suggestion, we have added the resistance information of the point contact to the revised manuscript:

1. Caption in Fig. 3. The PC resistance in the normal state is 10.7 Ω .
2. Caption in Supplementary Fig. 8. The PC resistance in the normal state is 4.0 Ω .
3. Caption in Supplementary Fig. 9. The PC resistance in the normal state is 15.3 Ω .
4. Caption in Supplementary Fig. 10. The PC resistance in the normal state is 7.2 Ω .
5. Caption in Supplementary Fig. 18 (Supplementary Fig. 17 in the last round of review). The resistance for PC1, PC2, and PC3 in the normal state are 4.9 Ω , 2.2 Ω , and 4.2 Ω , respectively.

In our point-contact experiments, PCS showing zero-bias conductance peak (ZBCP) are observed by using various metallic tips, including Au, PtIr, and Ni tips. Intriguingly, for three point contacts obtained by using Au tips, the temperature dependence of zero-

bias conductance height and the full width at half maximum (FWHM) of the ZBCP can be fitted well by the exponential function (Supplementary Fig. 17), consistent with the characteristic of the Majorana bound state [*Nano. Lett.* **19**, 4890-4896 (2019); *Nat. Phys.* **16**, 536–540 (2020)]. The reason why only gold tips can detect the ZBCP showing the feature of the Majorana bound state has not been fully understood, and it might be related to the material properties of gold tips, such as the work function. As we mentioned above, the main observation of our work is the emergent superconductivity in the TbMn₆Sn₆/metal heterostructure. Therefore, our point-contact experiments on TbMn₆Sn₆ are mainly focused on detecting superconducting signals to verify and support the electrical transport measurement results.

We thank the reviewer for the kind reminder “*this behavior cannot be explained by the mechanism of point contact in thermal regime ... (page 7, line 161 in SI), but no clear explanation or reference has not been provided*”. It is worth mentioning that the exponential temperature dependence of zero-bias conductance height and the FWHM of the ZBCP shown in Supplementary Fig. 18 (Supplementary Fig. 17 in the previous version) is not expected for the PCS in the thermal regime but has been reported as evidence of Majorana bound states [*Nat. Phys.* **16**, 536–540 (2020); *Nano. Lett.* **19**, 4890-4896 (2019)]. Furthermore, as the reviewer said in the previous comment, for the point contact in the thermal regime, “*the conductance spectra may reflect temperature dependence resistance*”. Here, we assume that the point contact in Supplementary Fig. 18 is in the thermal regime, and the ZBCP is triggered by Joule heating in the contact region. Taking the PC1 shown in Supplementary Fig. 18 as an example, we convert the voltage (V) to the temperature (T_{PC}) according to the formula [*J. Phys.: Condens. Matter* **10**, 8905 (1998)]: $T_{PC}^2 = T_{bath}^2 + V^2/4L$, where $L = 2.45 \times 10^{-8} \text{ V}^2/\text{K}^2$ is the Lorenz number. T_{bath} is bath temperature, for PC1 shown in Supplementary Fig. 18a, we take $T_{bath} = 1.1 \text{ K}$. We plot the converted curve of the resistance versus temperature from PCS (black) together with the curve of differential resistance versus temperature for PC1 (red) in the Fig. R2. As shown in Fig. R2, there is a notable difference between these two curves, indicating that the ZBCP in Supplementary Fig. 18a is not triggered by Joule heating. Hence, the point contacts shown in Supplementary Fig. 18 are far from the thermal regime.

Fig. R2 | The curve of the resistance versus temperature converted from PCS in Supplementary

Fig. 18a (black) and the curve of differential resistance versus temperature (red) measured on the same point contact. The notable difference between the two curves indicates the spectra and zero bias conductance peak shown in Supplementary Fig. 18a are not a result of Joule heating.

To avoid possible misunderstanding, we have revised the sentence “*this behavior cannot be explained by the mechanism of point contact in thermal regime ...*” mentioned by the reviewer as follows.

In the revised Supplementary Text VI: The exponential temperature dependence of zero-bias conductance height and the FWHM of the ZBCP is consistent with the characteristic of the Majorana bound state^{19,20} but not expected for the superconducting point contact in the thermal regime. Thus, the topological non-trivial superconductivity is a potential explanation for the PCS shown in Supplementary Fig. 18.

As for the PCS results with soft contacts mentioned by the reviewer, we would like to point out that no soft point-contact experiments with silver paste are involved in our work. The word ‘soft’ or ‘hard’ in our manuscript is used to describe the hardness of the tips. The ‘soft tip’ means ‘Au tip’, because the mineral hardness of Au is lower than other tip materials (PtIr, Ni). Note that the interface superconductivity in the TbMn₆Sn₆/metallic tip point contact is insensitive to the hardness of the tip, demonstrating that the interface superconductivity is not sensitive to local pressure. In the revised manuscript, we have added ‘(Au tip)’ after the word ‘soft’ and ‘(PtIr tip)’ after ‘hard’, to avoid possible misunderstanding.

In the revised manuscript (revised 2nd paragraph, page 8), we added ‘(Au tip)’ after ‘soft’, and ‘(PtIr tip)’ after ‘hard’.

Comment:

6) To validate their PCS results, they may try BTK fits to their dI/dV spectra. Is a superconducting gap estimated from the dI/dV spectra consistent with the other data?

Response

We thank the reviewer for this insightful suggestion. Following the suggestion, we tried to fit PCS by using the BTK model, a typical example is shown in Fig. R3a. The fitted superconducting gap value is 0.66 meV, which is approximately equal to the value estimated by the bias voltage of the conductance peak (~ 0.70 mV). However, in addition to the two conductance peaks, two conductance dips can also be observed in the spectrum, which deviates from the fitting line. In our experiments, the two conductance peaks in PCS are always followed by two conductance dips. If the point contact is in a ballistic regime, this type of PCS may indicate that the superconductivity is unconventional [*Phys. Rev. B* **85**, 024522 (2012); *Physica E* **55**, 25–29 (2014)]. In this case, it is hard to obtain reasonable fitting results without knowing the specific symmetry of the superconducting pairing. Alternatively, a point contact in the intermediate regime can also show two conductance peaks and two conductance dips in the PCS [*Supercond. Sci. Technol.* **23**, 043001 (2010); *Nat. Mater.* **15**, 32–37 (2016)], in which the energy value (meV) corresponding to the conductance peak can be roughly considered as the superconducting gap value when $T \ll T_c$. In Fig. R3b, we plot the bias voltage of conductance peaks in PCS obtained by Ni tips vs. T_c values. The T_c values are estimated from R - T curves of the corresponding superconducting state. As shown in Fig. R3b, a positive correlation between peak bias and T_c values is detected, demonstrating the PCS results are valid.

Fig. R3 | The analysis of the conductance peaks of PCS. **a.** The BTK fitting of one typical PCS, where Z is the barrier parameter, Γ is the broadening parameter, Δ_s is the fitting gap value and Re is extra resistance. **b.** The statistics of the conductance-peak bias voltages and the superconducting transition temperature T_c for the PC made on (001) surface of $TbMn_6S_6$ by using Ni tips. The T_c values are estimated from R - T curves of the corresponding superconducting states.

In the revised manuscript, we have added Fig. R3b as Supplementary Fig. 12.

1. Supplementary Fig. 12

Supplementary Fig. 12 | The statistics of the conductance-peak bias voltages and the superconducting transition temperature T_c for the point contacts made on the (001) surface of TbMn_6S_6 by using Ni tips. A positive correlation between peak bias and T_c values is detected.

Comment:

7) (Minor) In the previous comment 1, I pointed out that the zero-resistance is not observed. A simple explanation could be the inhomogeneity of the superconducting phase, but they need to discuss that in the manuscript. (Related to the response to Reviewer #2's comment ii) Is it possible to be related to the "partial" (non-superconducting) polycrystalline TbMn₆Sn₆ and the rest (superconducting but not-identified phase)?

Response

We thank the reviewer for the suggestion and comment. As the reviewer mentioned, the non-zero resistance could be due to the '*inhomogeneity of the superconducting phase*'. In the last round of review, we added related discussions to the manuscript (3rd paragraph, Page 6). Here, we list the content below.

'Due to the non-van der Waals nature of TbMn₆Sn₆ crystals, it is challenging to obtain a large area of atomic-level flat surfaces from the micrometer-scale TbMn₆Sn₆ single crystals. As a result, when depositing a metallic film onto the TbMn₆Sn₆ surface, it is difficult to achieve a uniform and consistent contact condition, which may lead to inhomogeneous superconductivity with non-zero residual resistance at low temperatures.'

As mentioned in the response to Comment 3, there are no other elements or compounds besides Sn that could exhibit superconductivity in our samples. Hence, no rest superconducting phase exists in the TbMn₆Sn₆/metal heterostructure.

Comment:

8) (Minor) In the abstract, “two-dimensional semiconductors proximity-coupled to *s*-wave superconductors and insulating ferromagnets” – this seems not to reflect their work.

Response

The sentence mentioned by the reviewer is a theoretical strategy for topological superconductivity in a heterostructure that contains an *s*-wave superconductor, a 2D semiconductor with Rashba SOC, and a ferromagnetic insulator (Fig. 5a). This proposal has not been experimentally realized yet. In our work, we show that the emergent superconductivity at the interface of topological kagome magnet/metal heterostructures meets all the theoretical criteria for generating topological superconductivity in a 2D system, including strong SOC, proximity-coupled 2D superconductivity, and ferromagnetism. Therefore, the TbMn₆Sn₆/metal heterostructure is similar to the theoretical strategy shown in Fig. 5a. More detailed discussions can be referred to the “Discussion” section and Fig. 5.

Comment:

9) (Minor) double conductance peaks combined with double conductance dips in the PCS for the case in the intermediate regime — this is the wrong expression. And the double conductance enhancement can be observed when $Z=0$ (no barrier)

Response

The “double conductance peaks” mentioned in the text refer to the two conductance peaks in the PCS, not the double conductance enhancement. For a point contact in the intermediate regime, the corresponding point contact spectrum often exhibits two conductance peaks and two conductance dips [*Supercond. Sci. Technol.* **23**, 043001 (2010); *Nat. Mater.* **15**, 32–37 (2016)]. The two conductance peaks are attributed to the Andreev reflection, which can reflect the superconducting gap information. The two conductance dips are related to the thermal effect and the critical current effect. To avoid possible confusion, we use “two” instead of “double” in the revised version.

The revised contents are listed below:

1. 1st paragraph, Page 7, The PCS at 1.5 K (marked in Fig. 3b) shows clearly two conductance peaks at ± 0.6 mV. For a PC, the two conductance peaks are usually taken as the hallmark of Andreev reflection processes at the interface between a normal metal and a superconductor³². With the increase of temperature or magnetic field, the two conductance peaks are gradually suppressed, consistent with the signature of weakening superconductivity, confirming the emergence of superconductivity at the TbMn₆Sn₆/PtIr tip PC interface.
2. Supplementary Text V: When point-contact Andreev-reflection measurements are conducted on a conventional superconductor, there would be only two conductance peaks in the point-contact spectra for the case in the ballistic regime, and two conductance peaks combined with two conductance dips in the PCS for the case in the intermediate regime.

Comment:

10) (Minor) In the response letter, I suggest they provide revised content to each comment if they have revised the manuscript according to the reviewers' comments.

Response

We thank the reviewer for the suggestion. In each response to the comments, we have listed the corresponding revised content (purple color) made to the manuscript.

Reviewers' Comments:

Reviewer #3:

Remarks to the Author:

The authors have diligently responded to the comments and questions raised by the reviewer, resulting in substantial improvements to the manuscript. Many of the concerns have been effectively addressed. The significant experimental effort and detailed discussion dedicated to the revised manuscript are worth reporting. Many questions still remain, but I hope that this work inspires further research not only from the authors themselves but also from the broader scientific community. I recommend that the authors add Figure R2 and Figure R3a to the supplementary information; these additional figures would be appreciated by the readers. Considering these substantial improvements and the overall quality of the research, I recommend the acceptance of this paper for publication.

1333

Response to Reviewers

1334

1335 The reviewers' original comments are shown in blue characters. Our responses are
1336 shown in black characters.

1337

1338

Reviewer #3 (Remarks to the Author):

1339

Comment:

1340 The authors have diligently responded to the comments and questions raised by the
1341 reviewer, resulting in substantial improvements to the manuscript. Many of the
1342 concerns have been effectively addressed. The significant experimental effort and
1343 detailed discussion dedicated to the revised manuscript are worth reporting. Many
1344 questions still remain, but I hope that this work inspires further research not only from
1345 the authors themselves but also from the broader scientific community. I recommend
1346 that the authors add Figure R2 and Figure R3a to the supplementary information; these
1347 additional figures would be appreciated by the readers. Considering these substantial
1348 improvements and the overall quality of the research, I recommend the acceptance of
1349 this paper for publication.

1350

Response

1351 We express our sincere appreciation to Reviewer #3 for recommending the publication
1352 of our manuscript in *Nature Communications*. As suggested by the reviewer, we have
1353 included Fig. R2 and Fig. R3a in the revised Supplementary Information.

1354

1355

1356

1357